# Flat-space Partial Waves From Conformal OPE Densities

**Balt C. van Rees**[1] **and Xiang Zhao**[1,2]

[1] CPHT, CNRS, École Polytechnique, Institut Polytechnique de Paris, Route de Saclay, 91128 Palaiseau, France

[2] Fields and String Laboratory, Institute of Physics, Ecole Polytechnique Fédérale de Lausanne (EPFL), Rte de la Sorge, CH-1015 Lausanne, Switzerland

`balt.van-rees@polytechnique.edu, xiang.zhao@epfl.ch`

## Abstract

We consider the behavior of the OPE density $c(\Delta, \ell)$ for conformal four-point functions in the flat-space limit where all scaling dimensions become large. We find evidence that the density reduces to the partial waves $f_\ell(s)$ of the corresponding scattering amplitude. The Euclidean inversion formula then reduces to the partial wave projection and the Lorentzian inversion formula to the Froissart-Gribov formula. The flat-space limit of the OPE density can however also diverge, and we delineate the domain in the complex $s$ plane where this happens. Finally we argue that the conformal dispersion relation reduces to an ordinary single-variable dispersion relation for scattering amplitudes.

## 1  Introduction

In the flat-space limit, the boundary correlation functions for a QFT in hyperbolic space should morph into its S-matrix elements. This idea dates back to the early days of the AdS/CFT correspondence [1–4]. In subsequent works several potential (and often related) prescriptions were put forward, see [5–11] for some non-perturbative discussions. More recently the flat-space limit for *gapped* non-gravitational theories in AdS also became a topic of interest [9, 12–19]. For the boundary correlation functions this corresponds to a limit where all scaling dimensions become large.

For this paper our starting point will be a precise conjecture from [13, 15]:

$$\langle \underline{\tilde{k}}_1 \dots \underline{\tilde{k}}_a | S | \underline{k}_1 \dots \underline{k}_b \rangle \overset{?}{=} \lim_{R \to \infty} \left( \sqrt{Z} \right)^{a+b} \langle \mathcal{O}(\tilde{n}_1) \dots \mathcal{O}(\tilde{n}_a) \mathcal{O}(n_1) \dots \mathcal{O}(n_b) \rangle |_{\text{S-matrix}} \tag{1.1}$$

We call this the *S-matrix conjecture*. We believe that (1.1) has significant potential to help us understand scattering amplitudes non-perturbatively. The right-hand side is a conformal correlation function, which is a well-understood object since it has conformal block decompositions that converge absolutely in a large domain. The left-hand side, on the other hand, is much more mysterious. In particular, it is notoriously difficult to rigorously derive analyticity properties of scattering amplitudes from the Wightman axioms and the LSZ prescriptions [20]. What can we learn if we use equation (1.1) instead?

In our earlier paper [21] we showed that relatively mild assumptions suffice to show that (1.1) leads to an amplitude that is analytic in a large domain and compatible with unitarity. In this paper we will investigate the flat-space limit of the OPE density $c(\Delta, \ell)$ and the Euclidean [22] and Lorentzian [23] 'inversion formulas' that compute it. Finally, we will also investigate the flat-space limit of the conformal dispersion relation [24].

A principal result of this paper is what we call the *partial wave conjecture*:

$$\boxed{n_\ell^{(d)} f_\ell(s) = \lim_{R\to\infty} \frac{c_{\mathrm{conn}}(\Delta,\ell)}{c_c(\Delta,0)}\bigg|_{\Delta=\frac{d}{2}+R\sqrt{s}}}$$

(1.2)

for some domain in the complex $s$ plane. In words, the OPE density of the *connected* correlator $c_{\mathrm{conn}}(\Delta,\ell)$, when divided by the spin-0 OPE density of the contact diagram $c_c(\Delta,0)$, should reduce to the partial wave $f_\ell(s)$ in the flat-space limit. Important is also the identification $\Delta = \frac{d}{2} + R\sqrt{s}$, which appeared earlier in other phase shift formula analyses that we discuss below. (The need to consider the OPE density of the connected correlation function is explained in subsection 4.2.)

Much like the analysis in [21], our expectation is that (1.2) will be useful to derive new non-perturbative results of the partial waves, in particular their analyticity properties. (Results obtained before 1970 on the analyticity of $f_\ell(s)$ in $s$ are reviewed in [25].) To do so we must however first understand the conjecture and its regime of validity, and this is the aim of the present paper.

Our main evidence for the partial wave conjecture (1.2) is three-fold and discussed in detail in the next four sections.

1. The first important verification arises from the various integral formulas that relate the position-space correlation function and the OPE density. The general idea here is invariably that a *saddle-point approximation* relates equation (1.2) to the S-matrix conjecture (or more precisely the related amplitude conjecture introduced in equation (2.3) below). If (!) this saddle point approximation is reliable one immediately finds that the Euclidean inversion formula becomes the partial wave projection (discussed in section 3), that the principal series decomposition of the correlator becomes the partial wave decomposition of the amplitude (section 8.1), and that the Lorentzian inversion formula becomes the Froissart-Gribov formula (section 5).

2. Our second verification (section 8) is that we can connect our partial wave formula to the hyperbolic partial waves introduced in [21] and thereby also to the phase shift formulas of [12, 15], again assuming (!) the reliability of a saddle point analysis. We note that this phase shift formula, which convolves the OPE data with a Gaussian function, makes sense only for physical kinematics, *i.e.* real $s$ above threshold. The partial wave conjecture (1.2) thus predicts how the phase shift formula can be extended to the complex $s$ plane.

3. Our third piece of evidence is that the partial wave conjecture works to some extent in several examples like the exchange diagrams (subsection 4.1 and section 7). Note that, unlike the previous two items, here we need not assume the validity of the S-matrix or amplitude conjecture in any sense because we can explicitly compute the relevant limits.

Unfortunately saddle points in the large-$\Delta$ limit of conformal correlators have a habit of not always being reliable. This happens because a saddle point is only guaranteed to be the dominant contribution if the integration contour lies sufficiently close to the steepest descent contour. But sometimes the integration contour is far away and, furthermore, there are certain non-analyticities in the integrand that prevent the integration contour from being deformed to the steepest descent contour. This leads to additional *contour contributions*, see figure 2 on page 21 for a pictorial representation. Generically these contour contributions are either subleading, so they can be ignored, or they are dominant so they entirely invalidate the saddle point approximation. It is these *divergent contour contributions* that spoil all our conjectures in some subset of their potential domain of validity.

In section 2 we review how such divergent contour contributions can invalidate the S-matrix conjecture for certain kinematics, which was discussed first in [15]. In Witten diagrams the divergent contour contributions can be given a physical meaning: they correspond to AdS Landau diagrams [15] where on-shell intermediate particles travel over distances much larger than the AdS radius.[1]

We undertake a general study of possible divergent contour contributions for the partial wave conjecture in section 6. We find that, even if nothing goes wrong with the S-matrix conjecture, the partial wave conjecture (1.2) *cannot* be everywhere valid: the limit unavoidably diverges in a subset of the complex $s$ plane. We will explain that the non-analyticity behind the divergent contour contribution is just the left cut of the partial waves. The partial wave conjecture then has a chance to work only in the complement of this region, at least without any further modifications where somehow these divergences are subtracted away.

The simplest perturbative example where the partial waves have a left cut is the $t$-channel exchange diagram, which we discuss in section 7. Here the amplitude conjecture can fail either on the primary sheet or on the secondary sheets, and we connect these divergences to a failure of the partial wave conjecture. In the end we obtain a slightly bigger region with divergences than in the more agnostic analysis of section 6.

The Lorentzian inversion formula is intimately related to the conformal dispersion relation [24]. In section 9 we therefore also analyze the fate of the latter in the flat-space limit. We show that it becomes an ordinary fixed-$t$ dispersion relation. Of course it is also plagued by various divergences, but these were essentially analyzed in our earlier paper [21].

The general picture that emerges is that there are a variety of prescriptions to extract scattering amplitudes from correlators in the flat-space limit. In practice one finds that all these prescriptions are related by some sort of saddle point approximation. The prescriptions however all have bad regions where the limit diverges, often due to divergent contour contributions. Delineating these regions is much less straightforward, and this is what necessitates the more intricate analyses below. In other words, the flat-space limit only looks easy if one recklessly interchanges limits and integrals.

---

[1]Perhaps these divergences may be subtracted away in a more refined analysis, as was done for example in [18, 21], but we leave such an analysis to future work.

## 2 Axiomatics of the flat-space limit

We will concern ourselves with conformal four-point functions of identical scalar boundary operators for a gapped QFT in $\text{AdS}_{d+1}$ with curvature radius $R$. By AdS covariance these obey all the usual axioms of CFT correlation functions, with the notable absence of a stress tensor among the operator spectrum. In this setup the state-operator correspondence is understood to be between the states in the QFT Hilbert space on the AdS cylinder and the local operators on the boundary, cf. the discussion in [12].

Although we will not use it explicitly in this paper, it is useful to briefly discuss the spectrum of the theory in the flat-space limit. We would naturally assume that the Hilbert space for the QFT in AdS approaches in some sense the Fock space of a gapped QFT in flat space, consisting of the vacuum plus single- and multi-particle states, just as is the case in a generalized free theory in AdS. Non-trivial conformal primaries correspond to particles in the center-of-mass frame, and conformal descendants to boosted particles. The scaling dimension of a primary operator corresponding to a single-particle state of mass $m$ is given by the familiar Casimir relation, for example for a scalar operator we have

$$\Delta(\Delta - d) = m^2 R^2 \,. \tag{2.1}$$

If there is a single-particle state with dimension $\Delta_{\mathcal{O}}$ then there should be multi-particle states with dimension approximately $k\Delta_{\mathcal{O}}$ plus integer shifts. For example, as is well-known from generalized free field theory, the two-particle states have quantum numbers

$$(\Delta, \ell) = (2\Delta_{\mathcal{O}} + 2n + \ell + \gamma_{n,\ell}, \ell), \qquad n \in \{0, 1, 2, \ldots\} \,, \tag{2.2}$$

with anomalous dimensions $\gamma_{n,\ell}(mR)/\Delta_{\mathcal{O}} \to 0$ as $R \to \infty$. In [12] these anomalous dimensions were connected to scattering phase shifts; see also [15] for some elaborations, but here we will not pursue this approach to the flat-space limit.

Let us now turn to correlation functions. We will consider general four-point functions of operators dual to single-particle states. For computational simplicity we will often restrict ourselves to identical operators, which yield elastic amplitudes in the flat-space limit, but we expect the results established below to hold more generally. We have not considered correlation functions with more than four operators, but it would be very interesting to do so in the future.

Our workhorse formula for the flat-space limit of the four-point function is a small refinement of equation (1.1). This is because the S-matrix element on the left-hand side contains a momentum-conserving delta function which we would like to strip away to obtain just the scattering amplitude. This can be done before taking the large $R$ limit by simply dividing the correlation function on the right-hand side by the contact diagram with unit coefficient. (The latter was indeed verified in [15] to reduce to the momentum-conserving delta function in the flat-space limit.) Denoting the contact diagram as $\mathcal{G}_c(\tilde{n}_1, \tilde{n}_2, n_3, n_4)$, this yields the

*amplitude conjecture* of [15]:

$$\mathcal{T}\left(\tilde{k}_1, \tilde{k}_2; k_3, k_4\right) \stackrel{?}{=} \lim_{R \to \infty} \left. \frac{\langle \mathcal{O}\left(\tilde{n}_1\right) \mathcal{O}\left(\tilde{n}_2\right) \mathcal{O}\left(n_3\right) \mathcal{O}\left(n_4\right) \rangle_{\text{conn}}}{\mathcal{G}_c\left(\tilde{n}_1, \tilde{n}_2, n_3, n_4\right)} \right|_{\text{S-matrix}} . \tag{2.3}$$

The advantage of this conjecture is that it also makes sense for unphysical (complex) momenta.

The 'S-matrix' configuration in equations (1.1) and (2.3) fixes the insertion points $n_i$ in terms of the $k_i$, see [15] for the precise map. For this paper we only need the corresponding relation between conformal cross ratios and Mandelstam invariants. This relation defines the *conformal Mandelstam invariants* as follows. Consider a four-point function $\mathcal{G}(z, \bar{z})$ where $z$ and $\bar{z}$ are the usual functions of the cross ratios $u$ and $v$. We then introduce the radial coordinates [26]

$$z = \frac{4\rho}{(1+\rho)^2}, \qquad\qquad \bar{z} = \frac{4\bar{\rho}}{(1+\bar{\rho})^2} , \tag{2.4}$$

and their polar decomposition $\rho = re^{i\phi}$ and $\bar{\rho} = re^{-i\phi}$, which implies

$$r = \sqrt{\rho\bar{\rho}}, \qquad\qquad \eta = \cos(\phi) = \frac{1}{2}\left(\sqrt{\frac{\rho}{\bar{\rho}}} + \sqrt{\frac{\bar{\rho}}{\rho}}\right) . \tag{2.5}$$

The conformal Mandelstam invariants that follow from the amplitude conjecture are then:

$$r(s) = \frac{2m - \sqrt{s}}{2m + \sqrt{s}} , \qquad\qquad \eta = \cos(\phi) = -1 + \frac{2t}{4m^2 - s} . \tag{2.6}$$

The Mandelstam variables $s$ and $t$ so defined (most conveniently with $m^2 = 1$) are also perfectly suitable for general conformal correlation functions, even before taking the flat-space limit. We will show their usefulness in several examples below. Notice that $\phi$ is also equal to the scattering angle in flat space, so $\eta$ here is the same as the $\eta$ in equation (3.1) below.

## 2.1 Divergences and subtractions

Somehwat disappointingly we know that the amplitude conjecture *cannot* work for all complex $s$ and $t$. This is easily observed at a perturbative level. For individual Witten diagrams there are associated *AdS Landau diagrams* [15] which for certain kinematics can give rise to divergences in the flat-space limit. Within these 'blobs' the limit is infinity and the amplitude conjecture simply does not hold.

As a pertinent example we can consider an $s$-channel Witten exchange diagram for a particle of mass $\mu$. For most values of the Mandelstam $s$, the flat-space limit is the expected

pole but sometimes we find a divergence. Following the notation of equation (2.3), we have:[2]

$$\lim_{R \to \infty} \left. \frac{\mathcal{G}_{\text{s-exch}}(n_i)}{\mathcal{G}_c(n_i)} \right|_{\text{S-matrix}} = \begin{cases} \infty & \text{if } s \in D_\mu \\ -\dfrac{1}{s - \mu^2} & \text{if } s \notin \overline{D_\mu} \end{cases} \tag{2.7}$$

If $\mu < 2m$ then $D_\mu$ is a disk-like region of the complex $s$ plane, entirely contained with the disk centered at $s = 4m^2$ with radius $4m^2 - \mu^2$, see figure 4(a) on page 25 for examples. The pole at $s = \mu^2$ is the leftmost point of $D_\mu$, and that part of $D_\mu$ necessarily extends into the physical region with $s > 4m^2$. On the first sheet $D_\mu$ is empty for $\mu > 2m$, but in section 7.2 we show that divergences do occur on the secondary sheets.

There are good indications that these divergences also persist beyond individual Witten diagrams to the non-perturbative level. One simple example would be the case where the non-perturbative flat-space amplitude has a bound state pole below threshold. Since the correlator (divided by the contact diagram) is analytic for all finite $R$, the limit must diverge at least at the location of the pole. Such a divergence can however not be limited to a single point. It follows from a simple application of Cauchy's theorem and Montel's theorem (as formulated in [27]) that the correlator cannot remain locally bounded in a small annulus *around* the pole. More generally, then, our expectation is that non-analyticities in the amplitude will be 'cloaked' in blobs where the flat-space limit diverges.

We made progress towards tackling these questions non-perturbatively in [21]. We used the Polyakov-Regge block decomposition [28–33] where conformal blocks are essentially replaced by Witten exchange diagrams (in two different channels). We then assumed (i) convergence of the Polyakov-Regge block decomposition for all finite $R$, (ii) a spectral gap where $\Delta > \sqrt{2}\Delta_{\mathcal{O}}$ for all $\Delta$ in the conformal block decomposition of $\langle \mathcal{O}\mathcal{O}\mathcal{O}\mathcal{O} \rangle$, (iii) a pointwise finite limit in a region called $E'$ defined by $s, t, u < 2m^2$ which is a subset of the Euclidean Mandelstam triangle.

Our main structural result is then that, for certain fixed $u$, a suitably modified sum over Polyakov-Regge blocks cannot diverge anywhere in the complex $s$ plane. The putative divergences arising from the Polyakov-Regge blocks with $\mu < 2m$ and their images under crossing can be subtracted away, and afterwards one finds analyticity and polynomial boundedness in the complex $s$-plane outside of the cut.

A second result of [21] is that the amplitude for physical kinematics cannot diverge either. This arises by virtue of a convergent expansion in *hyperbolic partial waves* which satisfy the non-linear unitarity inequality and the finiteness of their sum in the forward limit. The details of these derivations are reviewed in appendix C.

---

[2] Here we ignore the boundary of $D_\mu$, which contains points where the limit does not exist because of infinite oscillations.

# 3 Partial waves from the Euclidean inversion formula

In this section we will use the Euclidean inversion formula to derive, from a saddle point analysis, the partial wave conjecture of equation (1.2). In order to do so we assume validity of the amplitude conjecture (2.3) at least for some values of the Mandelstam invariants.

## 3.1 Partial wave conventions

We collect here the main equations concerning the partial wave decomposition of a two-to-two scattering amplitude and the associated unitarity equations.

In terms of[3]

$$t(\eta) := -\frac{1}{2}(s - 4m^2)(\eta + 1) \tag{3.1}$$

the partial wave decomposition reads

$$\mathcal{T}(s, t(\eta)) = \sum_{\ell=0}^{\infty} n_\ell^{(d)} f_\ell(s) P_\ell^{(d)}(\eta), \tag{3.2}$$

where $P_\ell^{(d)}$ is proportional to the Gegenbauer polynomial $C_\ell^{(d)}(\eta)$

$$P_\ell^{(d)}(\eta) = {}_2F_1\left(-\ell, \ell + d - 2, \frac{d-1}{2}, \frac{1-\eta}{2}\right) = \frac{\ell!}{(d-2)_\ell} C_\ell^{(d)}(\eta). \tag{3.3}$$

We will be using the conventions spelled out in [34] in which the normalization factor equals[4]

$$n_\ell^{(d)} = \frac{2^{d+1} \pi^{\frac{d-1}{2}} (2\ell + d - 2) \Gamma(\ell + d - 2)}{\Gamma\left(\frac{d-1}{2}\right) \Gamma(\ell + 1)}. \tag{3.4}$$

Using the orthogonality relation

$$\frac{1}{2} \int_{-1}^{1} d\eta \left(1 - \eta^2\right)^{\frac{d-3}{2}} P_\ell^{(d)}(\eta) P_{\ell'}^{(d)}(\eta) = \frac{\delta_{\ell\ell'}}{\mathcal{N}_d n_\ell^{(d)}}, \quad \mathcal{N}_d = \frac{(16\pi)^{\frac{1-d}{2}}}{2\Gamma\left(\frac{d-1}{2}\right)}, \tag{3.5}$$

we find that the inverse of (3.2) reads

$$f_\ell(s) = \frac{\mathcal{N}_d}{2} \int_{-1}^{1} d\eta (1 - \eta^2)^{\frac{d-3}{2}} P_\ell^{(d)}(\eta) \mathcal{T}(s, t(s, \eta)). \tag{3.6}$$

If we now define the phase shift as

$$e^{2i\delta_\ell(s)} := 1 + i \frac{(s - 4m^2)^{\frac{d-2}{2}}}{\sqrt{s}} f_\ell(s) \tag{3.7}$$

---

[3]Note that in this paper we use the definition $t := -(k_1 + k_4)^2$ to be consistent with the $t$-channel OPE limit in the CFT language, in which operator 2 approaches operator 3. This is why the $t = 0$, which usually is the forward limit, here corresponds to $\eta = -1$ instead.

[4]Note that we use $d$ for the spacetime dimension of the conformal theory and therefore $d_{\text{there}} = d_{\text{here}} + 1$.

then the unitarity condition reads

$$|e^{2i\delta_\ell(s)}| \leq 1 \tag{3.8}$$

and should hold for all physical $s \geq 4m^2$ and all (even) spins.

## 3.2 The principal series representation and its inverse

Consider a conformal four-point correlation function of scalar operators

$$\mathcal{G}(x_i) = T^{\Delta_i}(x_i)\mathcal{G}(z, \bar{z}) \tag{3.9}$$

with a kinematical prefactor of the form

$$T^{\Delta_i}(x_i) = \frac{1}{x_{12}^{\Delta_1+\Delta_2} x_{34}^{\Delta_3+\Delta_4}} \left(\frac{x_{23}}{x_{13}}\right)^{\Delta_{12}} \left(\frac{x_{24}}{x_{23}}\right)^{\Delta_{34}}, \qquad x_{ij} \equiv |x_i - x_j|, \tag{3.10}$$

In [22] it was suggested that this correlator can be written in the principal series representation as

$$
\begin{aligned}
\mathcal{G}(x_i) &= T^{\Delta_i}(x_i) \left( \delta_{12}\delta_{34} + \sum_{\ell=0}^{\infty} \int_{d/2}^{d/2+i\infty} \frac{d\Delta}{2\pi i} \rho(\Delta, \ell) \Psi_{\Delta,\ell}(z, \bar{z}) \right) \\
&= T^{\Delta_i}(x_i) \left( \delta_{12}\delta_{34} + \sum_{\ell=0}^{\infty} \int_{d/2-i\infty}^{d/2+i\infty} \frac{d\Delta}{2\pi i} \rho(\Delta, \ell) K_{\tilde{\Delta},\ell}^{\Delta_{34}} G_{\Delta,\ell}(z, \bar{z}) \right),
\end{aligned} \tag{3.11}
$$

where the conformal partial wave reads

$$\Psi_{\Delta,\ell}(z, \bar{z}) = \frac{1}{2} \left( K_{\tilde{\Delta},\ell}^{\Delta_{34}} G_{\Delta,\ell}(z, \bar{z}) + K_{\Delta,\ell}^{\Delta_{12}} G_{\tilde{\Delta},\ell}(z, \bar{z}) \right), \tag{3.12}$$

and

$$K_{\Delta,\ell}^{\Delta_{12}} = \left(-\frac{1}{2}\right)^{\ell} \frac{\pi^{\frac{d}{2}} \Gamma(\Delta - h) \Gamma(\Delta + \ell - 1) \Gamma\left(\frac{\tilde{\Delta}+\ell+\Delta_{12}}{2}\right) \Gamma\left(\frac{\tilde{\Delta}+\ell-\Delta_{12}}{2}\right)}{\Gamma(\Delta - 1) \Gamma(d - \Delta + \ell) \Gamma\left(\frac{\Delta+\ell+\Delta_{12}}{2}\right) \Gamma\left(\frac{\Delta+\ell-\Delta_{12}}{2}\right)}, \tag{3.13}$$

with $\tilde{\Delta} \equiv d - \Delta$ and $\Delta_{ij} \equiv \Delta_i - \Delta_j$. (We take $\Delta_{12} = \Delta_{34} = 0$ in the bulk of this paper.)

A decomposition as in (3.11) is possible because of the completeness of the partial waves. The density $\rho(\Delta, \ell)$ can be obtained from the inverse equation:

$$\rho(\Delta, \ell) = N(\Delta, \ell) \int d^2 z \, \mu(z, \bar{z}) \Psi_{\Delta,\ell}(z, \bar{z}) \left(\mathcal{G}(z, \bar{z}) - \delta_{12}\delta_{34}\right) \tag{3.14}$$

with the measure

$$\mu(z, \bar{z}) = \left|\frac{z - \bar{z}}{z\bar{z}}\right|^{d-2} \frac{((1 - z)(1 - \bar{z}))^{(\Delta_{21}+\Delta_{34})/2}}{(z\bar{z})^2} \tag{3.15}$$

and a rather unsightly normalization factor. Its full form can for example be found in [23],

but in the case $\Delta_{12} = \Delta_{34} = 0$ it reads:

$$N(\Delta, \ell) =$$
$$\frac{\pi^{-d-1} 2^{2\ell-1} \Gamma\left(\frac{d-2}{2}\right)^2 \Gamma(\Delta-1) \Gamma(d-\Delta-1) \Gamma\left(\frac{d}{2}+\ell\right) \Gamma(d+\ell-2) \Gamma(\ell+\Delta) \Gamma(d+\ell-\Delta)}{\Gamma(d-2)^2 \Gamma(\ell+1) \Gamma\left(\frac{d}{2}-\Delta\right) \Gamma\left(\Delta-\frac{d}{2}\right) \Gamma\left(\frac{d}{2}+\ell-1\right) \Gamma(\ell+\Delta-1) \Gamma(d+\ell-\Delta-1)} . \tag{3.16}$$

Notice that it is shadow symmetric, $N(\Delta, \ell) = N(d-\Delta, \ell)$, and therefore the density $\rho(\Delta, \ell)$ obtained from (3.14) is as well.

### 3.2.1 The contact diagram

By virtue of its appearance in the denominator of equation (2.3), the contact diagram plays a special role in our prescriptions. So let us note that the principal series representation of the contact diagram is given by:

$$\prod_{j=1}^{4} \mathcal{C}_{\Delta_j}^{\frac{1}{2}} \mathcal{G}_c(x_i) = T^{\Delta_i}(x_i) \int_{-\infty}^{\infty} \frac{d\nu}{2\pi} \underbrace{\frac{4\nu^2}{R^{d-3}} f_{\Delta_1 \Delta_2 \left(\frac{d}{2}+i\nu\right)} f_{\Delta_3 \Delta_4 \left(\frac{d}{2}-i\nu\right)} K_{\frac{d}{2}-i\nu,0}^{\Delta_{34}}}_{c_c\left(\frac{d}{2}+i\nu,0\right)} G_{\frac{d}{2}+i\nu,0}(z,\bar{z}), \quad (3.17)$$

where $f_{\Delta_i \Delta_j \Delta_k}$ is the OPE coefficient

$$f_{\Delta_i \Delta_j \Delta_k} = \frac{\pi^h}{2} \frac{\mathcal{C}_{\Delta_i} \mathcal{C}_{\Delta_j} \mathcal{C}_{\Delta_k}}{\Gamma(\Delta_i) \Gamma(\Delta_j) \Gamma(\Delta_k)} \Gamma(\Delta_{ijk}/2) \Gamma(\Delta_{ij\tilde{k}}/2) \Gamma(\Delta_{jki}/2) \Gamma(\Delta_{kij}/2), \tag{3.18}$$

with $\Delta_{ijk} \equiv \Delta_i + \Delta_j - \Delta_k$. The factor

$$\mathcal{C}_\Delta = \frac{\Gamma(\Delta)}{2\pi^h \Gamma(\Delta-h+1)} \tag{3.19}$$

is the normalization factor for the bulk-boundary propagator, which in embedding space reads:

$$G_{B\partial}^\Delta(X, P) = \frac{\mathcal{C}_\Delta}{R^{(d-1)/2}(-2P \cdot X/R)^\Delta}. \tag{3.20}$$

This representation of the contact diagram can be derived by making use of the spectral representation of the Dirac delta function and this is explained in appendix A.

For equal external dimensions $\Delta_i = \Delta_\mathcal{O}$, we can read off the OPE density as

$$c_c(\Delta, 0) = -\frac{\pi^{-\frac{3d}{2}} (d-2\Delta)^2 R^{3-d} \Gamma\left(\frac{\Delta}{2}\right)^4 \Gamma\left(\frac{d}{2}-\Delta\right) \Gamma\left(\Delta_\mathcal{O}-\frac{\Delta}{2}\right)^2 \Gamma\left(-\frac{d}{2}+\frac{\Delta}{2}+\Delta_\mathcal{O}\right)^2}{256 \Gamma(\Delta) \Gamma\left(\frac{d}{2}-\Delta+1\right) \Gamma\left(-\frac{d}{2}+\Delta+1\right) \Gamma\left(-\frac{d}{2}+\Delta_\mathcal{O}+1\right)^4} . \tag{3.21}$$

## 3.3 The naive flat-space limit

In this subsection we will show how a saddle-point approximation to the equations (3.14) and (3.11) produces the partial wave decomposition of the scattering amplitude if we assume the amplitude conjecture (2.3) to hold.

For the following it will be useful to define $\mathcal{T}_R(z, \bar{z})$ as the connected correlator divided by the contact diagram, i.e. for any finite $R$ we write:

$$\mathcal{G}_{\text{conn}}(x_i) = \mathcal{G}_c(x_i)\mathcal{T}_R(z, \bar{z}). \tag{3.22}$$

The amplitude conjecture then simply states that $\lim_{R \to \infty} \mathcal{T}_R(z, \bar{z})$ remains finite and equals the scattering amplitude $\mathcal{T}(s, t)$.

We substitute (3.22) into (3.14) and transform to the $(r, \phi)$ coordinates given in equation (2.5). We obtain:

$$\rho(\Delta, \ell) = N(\Delta, \ell) \int_0^1 dr \int_0^{2\pi} d\phi \, \mu(r, \phi)\Psi_{\Delta, \ell}(r, \phi)\mathcal{G}_c(r, \phi)\mathcal{T}_R(r, \phi) \tag{3.23}$$

where

$$\mu(r, \phi) = 2^{-d-1}r^{2(1-d)}\left(1 - r^2\right)\left(r^2 + 2r\cos(\phi) + 1\right)\left(r^2 - 2r\cos(\phi) + 1\right)|\sin(\phi)|^{d-2} \tag{3.24}$$

is the measure in these coordinates, including the Jacobian from the change of variables.

Now we take the flat-space limit of each ingredient. The large $\Delta_{\mathcal{O}}$ limit of the contact diagram in position space [15] reads:

$$\mathcal{G}_c(r, \phi) \overset{R \to \infty}{\longrightarrow} \frac{w_c r^{-1/2}(r+1)^3}{\sqrt{(r^2 + 2r\cos(\phi) + 1)(r^2 - 2r\cos(\phi) + 1)}}\left(\frac{16r^2}{(r+1)^4}\right)^{\Delta_{\mathcal{O}}} \tag{3.25}$$

with the normalization

$$w_c = 2^{-\frac{d+7}{2}}\pi^{-\frac{d-1}{2}}\Delta_{\mathcal{O}}^{\frac{d-5}{2}}R^{3-d}, \tag{3.26}$$

which was verified in [15] to be precisely the right one to reproduce the momentum-conserving delta function in the flat-space limit.[5]

The large $\Delta$ limit of the conformal block in turn is given by [35, 36]:

$$G_{\Delta, \ell}(r, \phi) \overset{\Delta \to \infty}{\longrightarrow} \frac{(4r)^\Delta P_\ell^{(d)}(\cos(\phi))}{(1 - r^2)^{(d-2)/2}\sqrt{(r^2 + 2r\cos(\phi) + 1)(r^2 - 2r\cos(\phi) + 1)}}. \tag{3.27}$$

---

[5]Since we work with unit normalized operators, this factor is different from the one quoted in [15] by an extra $\mathcal{C}_\Delta^{-2}$, which becomes $4\pi^d\Delta^{2-d}$ for large $\Delta$.

Here we use the following normalisation for the conformal blocks in $d$ dimension

$$G_{\Delta,\ell}(z,\bar{z}) \to \frac{(\frac{d-2}{2})_\ell}{(d-2)_\ell} z^{\frac{\Delta-\ell}{2}} \bar{z}^{\frac{\Delta+\ell}{2}} \qquad (z \ll \bar{z} \ll 1). \qquad (3.28)$$

The function $\Psi_{\Delta,\ell}$ is the sum of a conformal block and a shadow conformal block. Altogether we therefore obtain:

$$\rho(\Delta,\ell) = N(\Delta,\ell) K_{\Delta,\ell}^0 \pi^{\frac{1}{2}-\frac{3d}{2}} \Delta_{\mathcal{O}}^{\frac{3d}{2}-\frac{9}{2}} R^{3-d} 2^{-\frac{3d}{2}+2\Delta+4\Delta_{\mathcal{O}}-\frac{17}{2}} \times$$

$$\int_0^1 dr (1-r)^{\frac{d}{2}-1}(1+r)^{\frac{d}{2}-4\Delta_{\mathcal{O}}+2} r^{-d+\Delta+2\Delta_{\mathcal{O}}-\frac{3}{2}} \int_0^{2\pi} d\phi \, |\sin(\phi)|^{d-2} P_\ell^{(d)}(\cos(\phi)) \mathcal{T}_R(r,\phi)$$

$$+ (\Delta \to (d-\Delta)) \quad (3.29)$$

Assuming that $\mathcal{T}_R(r,\phi)$ remains finite, we can approximate the $r$ integrals for large $|\Delta|$ and $\Delta_{\mathcal{O}}$ by their saddle point. Depending on the sign of the real part of $\Delta$ one of the two terms will have a dominant saddle point. We will take $\mathrm{Re}(\Delta) > d/2$ and then it is actually the second term that dominates, with the saddle point located at

$$r_* = \frac{2\Delta_{\mathcal{O}} - \Delta}{2\Delta_{\mathcal{O}} + \Delta}. \qquad (3.30)$$

We will discuss below that the saddle point cannot always be trusted, but whenever it can we find that:

$$\frac{\rho(\Delta,\ell)}{\rho_c(\Delta,0)} \xrightarrow{R\to\infty} \frac{N(\Delta,\ell) K_{\Delta,\ell}^0}{N(\Delta,0) K_{\Delta,0}^0} \times \frac{\Gamma(\frac{d}{2})}{2\sqrt{\pi}\Gamma(\frac{d-1}{2})} \int_0^{2\pi} d\phi \, |\sin(\phi)|^{d-2} P_\ell^{(d)}(\cos(\phi)) \mathcal{T}_R(r_*,\phi). \quad (3.31)$$

where $\rho_c(\Delta,0)$ is the density for the contact diagram (which is only non-zero at spin 0). The above equation is very similar to the partial wave projection of a scattering amplitude given in (3.6). From matching the two sides we can infer the following two results. The location of the saddle point, together with $\Delta_{\mathcal{O}} = mR$ for the external scaling dimension, yields

$$\boxed{\Delta = \frac{d}{2} + R\sqrt{s}} \qquad (3.32)$$

and the partial waves themselves are then obtained from

$$f_\ell(s) = \lim_{R\to\infty} \mathcal{N}(\Delta,\ell) \frac{\rho_{\mathrm{conn}}(\Delta,\ell)}{\rho_c(\Delta,0)} \bigg|_{\Delta=d/2+R\sqrt{s}} \qquad (3.33)$$

with an overall normalization constant

$$\mathcal{N}(\Delta,\ell) = \frac{1}{2} \times \frac{(16\pi)^{\frac{1-d}{2}}}{\Gamma(\frac{d-1}{2})} \times \frac{N(\Delta,0) K_{\Delta,0}^0}{N(\Delta,\ell) K_{\Delta,\ell}^0} \times \frac{2\sqrt{\pi}\Gamma(\frac{d-1}{2})}{\Gamma(\frac{d}{2})} \qquad (3.34)$$

### 3.3.1 Changing normalization conventions

Things look a little bit nicer in the conventions used in [23]. Compared to that paper, we normalize our conformal blocks differently as $G^{(\text{here})} = G^{(\text{there})}(\frac{d-2}{2})_\ell/(d-2)_\ell$. The density $c(\Delta, \ell)$ used in [23] is then related to our $\rho(\Delta, \ell)$ as $c(\Delta, \ell) = \rho(\Delta, \ell)K_{\Delta,\ell}^{\Delta_{34}}(\frac{d-2}{2})_\ell/(d-2)_\ell$. In those conventions we simply find

$$n_\ell^{(d)} f_\ell(s) \overset{?}{=} \lim_{R\to\infty} \frac{c_{\text{conn}}(\frac{d}{2} + R\sqrt{s}, \ell)}{c_c(\frac{d}{2} + R\sqrt{s}, 0)} \tag{3.35}$$

where the normalization factor $n_\ell^{(d)}$ is the same in equation (3.4), which appeared in the partial wave decomposition (3.2) of the scattering amplitude. Equation (3.35) is exactly our partial wave conjecture (1.2) from the introduction.

## 4 Examples and discussions

In this subsection we consider two pertinent examples of the partial wave conjecture: an $s$-channel exchange Witten diagram and a disconnected diagram. For these examples we know $c(\Delta, \ell)$ exactly and therefore we can also check the validity of the partial wave conjecture outside of the regime of validity of the Euclidean inversion formula. We will see that both examples offer interesting non-perturbative lessons.

### 4.1 The $s$-channel scalar exchange Witten diagram and blobs

Verifying the validity of the partial wave conjecture (3.35) for an $s$-channel scalar exchange Witten diagram (with exchanged dimension $\Delta_b$) is a triviality. In appendix A we recall that the OPE density for the diagram is simply:

$$c_{s\text{-exch}}(\Delta, 0) = \frac{-R^2}{(\Delta - d/2)^2 - (\Delta_b - d/2)^2} c_c(\Delta, 0) \tag{4.1}$$

so, with $\mu = \Delta_b/\Delta$, the limit in (3.35) is just

$$\lim_{R\to\infty} \frac{c_{s\text{-exch}}(\frac{d}{2} + R\sqrt{s}, 0)}{c_c(\frac{d}{2} + R\sqrt{s}, 0)} = \frac{-1}{s - \mu^2}. \tag{4.2}$$

which perfectly matches the flat-space answer for an amplitude equalling $-1/(s - \mu^2)$.

It is interesting that we obtain the right answer for all complex $\Delta$. After all, as we wrote in equation (2.7), the diagrams with $\Delta_b < 2\Delta_\mathcal{O}$ have a non-empty 'blob' $D_\mu$ in the complex $s$ plane inside of which the flat-space limit diverges. However there is no such blob in equation (4.2), so in this sense the partial wave conjecture has a slightly *bigger* region of validity than the amplitude conjecture.

Let us now interpret the disappearance of the blobs from the viewpoint of the Euclidean inversion formula. First, recall that the finite-$R$ Euclidean inversion formula converges only

in the strip

$$d - \Delta_b < \mathrm{Re}(\Delta) < \Delta_b. \tag{4.3}$$

Here we use $\Delta_b$ as in the preceding paragraph, but our discussion is also valid non-perturbatively if we think of $\Delta_b$ simply as the first non-trivial operator in the OPE. In the previous section we used a saddle point analysis for the Euclidean inversion formula. How can it be insensitive to the blobs?

The first thing to note is that the saddle point itself cannot lie in the blob without violating (4.3), so without violating the validity of the analysis in the first place. Instead we can only reach the dangerous region in the complex $s$ plane (for the conformal partial wave) via an analytic continuation from inside the strip (4.3). This is why the divergence inside the blobs in the amplitude conjecture does not necessarily imply a corresponding problem with the partial wave conjecture.

The second part of the analysis pertains to the integration contour itself. The endpoint of the integration contour is at $r = 0$ which corresponds to $s = 4m^2$ and which definitely lies inside any $s$-channel blob that may be present. This however does not automatically lead to a divergence, since the integrand is exponentially suppressed away from the saddle point. From the absence of a divergence in the exact answer we conclude that this exponential suppression is sufficient to overcome the divergence of the correlator. The saddle point approximation is therefore reliable despite the existence of the blob.

### 4.2 Disconnected diagram

It is also of interest to study the disconnected pieces. With the aid of the Lorentzian inversion formula [23] one easily finds that the OPE density for the $t$-channel identity reads [37]

$$
\begin{aligned}
c_{\text{t-id}}(\Delta, \ell) =& \frac{\Gamma(\Delta - 1)\Gamma\left(\frac{\ell+\Delta}{2}\right)^4}{4\pi^2\Gamma\left(\Delta - \frac{d}{2}\right)\Gamma(l + \Delta - 1)\Gamma(l + \Delta)} \times \pi^2 \frac{\Gamma\left(\ell + \frac{d}{2}\right)\Gamma\left(\frac{d}{2} - \Delta_{\mathcal{O}}\right)^2}{\Gamma(\ell + 1)\Gamma\left(\Delta_{\mathcal{O}}\right)^2} \\
& \times \frac{\Gamma(\Delta + \ell)\Gamma\left(\frac{-\Delta + 2\Delta_{\mathcal{O}} + \ell}{2}\right)}{\Gamma\left(\frac{\Delta+\ell}{2}\right)^2\Gamma\left(\frac{\Delta+\ell}{2} - \Delta_{\mathcal{O}} + \frac{d}{2}\right)} \times \frac{\Gamma(\tilde{\Delta} + \ell)\Gamma\left(\frac{-\tilde{\Delta} + 2\Delta_{\mathcal{O}} + \ell}{2}\right)}{\Gamma\left(\frac{\tilde{\Delta}+\ell}{2}\right)^2\Gamma\left(\frac{\tilde{\Delta}+\ell}{2} - \Delta_{\mathcal{O}} + \frac{d}{2}\right)},
\end{aligned} \tag{4.4}
$$

where $\tilde{\Delta} \equiv d - \Delta$. Even for the relatively safe region $0 < s < 4m^2$, one finds that the limit

$$\lim_{\Delta \to \infty} \frac{c_{\text{t-id}}(\frac{d}{2} + R\sqrt{s}, \ell)}{c_c(\frac{d}{2} + R\sqrt{s}, 0)} \tag{4.5}$$

does not exist. The source of the issue lies in the Gamma functions with large *negative* argument. The cleanest way to capture the issue is to note that the limit exists if we factor out a trigonometric prefactor:

$$\lim_{\Delta \to \infty} (\text{ugly}) \frac{c_{\text{t-id}}(\frac{d}{2} + R\sqrt{s}, \ell)}{c_c(\frac{d}{2} + R\sqrt{s}, 0)} = n_\ell^{(d)} \frac{\sqrt{s}}{2(4m^2 - s)^{\frac{d-2}{2}}} \tag{4.6}$$

with

$$\text{ugly} = \frac{2\sin^2\left(\frac{\pi}{2}(d - 2\Delta_\mathcal{O})\right)}{\tan\left(\frac{\pi}{2}(d - \Delta + l)\right)\sin\left(\frac{\pi}{2}(2d - \Delta - 2\Delta_\mathcal{O} + l)\right)\sin\left(\frac{\pi}{2}(d + \Delta - 2\Delta_\mathcal{O} + l)\right)} . \tag{4.7}$$

Note that, apart from 'ugly', the limit is the reciprocal from the prefactor in the definition of the phase shift, see equation (3.7). (The apparent factor 2 mismatch disappears once we add the disconnected diagram in the other channel.) However even for real $\Delta$ the prefactor oscillates infinitely rapidly in the limit, whereas for complex $\Delta$ there are exponential divergences and the limit is certainly not finite.

Which lesson should we draw from this? As written the partial wave conjecture makes sense at most for (slightly) complex $s$ where there are no infinite oscillations due to the poles in $c_{\text{conn}}(\Delta, \ell)$. If we want to make sense of it for strictly real $s$ then it will presumably be necessary to average out these oscillations in some sense. Such an averaging procedure is in fact already used in the phase shift formula of [12], which works directly with the OPE data and only works for real $s$ above threshold. In section 8 show that the two formulas are complementary in the sense that the partial wave conjecture offers a complexification of the phase shift formula.

Returning now to the above analysis of the $t$-channel identity diagram, we see that this complexification does not work for the disconnected pieces: here the limit makes sense at most for real and physical kinematics, with some averaging over the OPE data as described by the phase shift formula (or the hyperbolic partial waves discussed in [21]). It is therefore essential to apply the partial wave conjecture to the *connected* correlator only, as we wrote in equation (1.2).

# 5   Partial waves from the Lorentzian inversion formula

In section 3.3 we obtained the partial wave conjecture (3.35) from a saddle-point approximation to the Euclidean inversion formula. Both sides of equation (3.35) can however also be calculated through an integral formula that has some analyticity in spin, namely the Froissart-Gribov formula and the Lorentzian inversion formula of Caron-Huot [23]. It is therefore natural to expect that these two integral formulae are also related through the flat-space limit. In this section we give a saddle-point argument for this. For classical limits of the Lorentzian inversion formula see [38].

**The Froissart-Gribov formula**

Let us first review the Froissart-Gribov formula. We will again consider elastic scattering of identical particles. For the (even-spin) flat-space partial waves we then find

$$f_\ell(s) = \frac{2\mathcal{N}_d}{\pi} \int_{t(\eta_0)=4m^2}^\infty d\eta(\eta^2 - 1)^{\frac{d-3}{2}} Q_\ell^{(d)}(\eta) \, \text{Disc}_t\left[\mathcal{T}(s, t(\eta))\right] \tag{5.1}$$

with the discontinuity defined as

$$\text{Disc}_y[\mathcal{T}(x,y)] := \frac{1}{2i}\left(\mathcal{T}(x,y+i\epsilon) - \mathcal{T}(x,y-i\epsilon)\right) \tag{5.2}$$

and where $\eta$ is the cosine of the scattering angle (2.6) and the Gegenbauer $Q$ function is[6]

$$Q_\ell^{(d)}(\eta) = \frac{c_\ell^{(d)}}{\eta^{\ell+d-2}}\, {}_2F_1\left(\frac{\ell+d-2}{2}, \frac{\ell+d-1}{2}, \ell+\frac{d}{2}, \frac{1}{\eta^2}\right), \quad c_\ell^{(d)} = \frac{\sqrt{\pi}\Gamma(\ell+1)\Gamma\left(\frac{d-1}{2}\right)}{2^{\ell+1}\Gamma\left(\ell+\frac{d}{2}\right)}, \tag{5.3}$$

which has a branch cut at $\eta \in [-1,1]$.

**The Lorentzian inversion formula**

Our aim will be to reproduce the Froissart-Gribov formula from the Lorentzian inversion formula [23] for the connected correlator, which reads

$$c_{\text{conn}}(\Delta,\ell) = 4 \times \frac{\kappa_{\Delta+\ell}}{4}\int_0^1 d\rho \int_\rho^1 d\bar\rho\, \mu(\rho,\bar\rho) G_{\ell+d-1,\Delta-d+1}(\rho,\bar\rho)\, \text{dDisc}_t\left[\mathcal{G}_{\text{conn}}(\rho,\bar\rho)\right], \tag{5.4}$$

where we used the $\rho, \bar\rho$ variables introduced in (2.4). Note the appearance of a factor of 4 in (5.4), which follows from combining $t$ and $u$-channel contribution and from restricting the integration region to $\bar\rho > \rho$. The double discontinuity is defined as

$$\text{dDisc}_t[g(\rho,\bar\rho)] := g(\rho,\bar\rho) - \frac{1}{2}g(\rho,1/\bar\rho - i\epsilon) - \frac{1}{2}g(\rho,1/\bar\rho + i\epsilon), \tag{5.5}$$

and the integration measure in these variables is

$$\mu(\rho,\bar\rho) = \frac{\left(1-\rho^2\right)\left(1-\bar\rho^2\right)}{16\rho^2\bar\rho^2}\left|\frac{(1-\rho\bar\rho)(\bar\rho-\rho)}{4\rho\bar\rho}\right|^{d-2}. \tag{5.6}$$

**The integration domain**

Before taking the flat-space limit it will be useful to take stock of the integration region. This is sketched in figure 1 in various coordinates.

The integral in (5.4) is such that $0 < \rho < \bar\rho < 1$, but the double discontinuity (5.5) in the integrand means that we evaluate the correlator $g(\rho,\bar\rho)$ *also* over a domain where $0 < \rho < 1 < \bar\rho < 1/\rho$. Altogether this gives the dark and light blue regions in figure 1(a). Notice that in this picture we cannot see the Euclidean domain where $\rho \in \mathbb{C}$ and $\bar\rho = \rho^*$.

For the saddle point analysis below we will transform to the $(r,\eta)$ coordinates, with $\eta = \cos(\phi)$. In that case the integration region is as shown in the figure 1(b). We note that it lies entirely above the line $\eta = 1$, corresponding to unphysical (Euclidean) angles. The line $\bar\rho = 1$ translates to $1 + r^2 - 2r\eta = 0$, and in these coordinates the Euclidean region is visible as the orange domain where $-1 < \cos(\phi) < 1$.

---

[6]We again use the convention of [34] and $d_{\text{there}} = d_{\text{here}} + 1$.

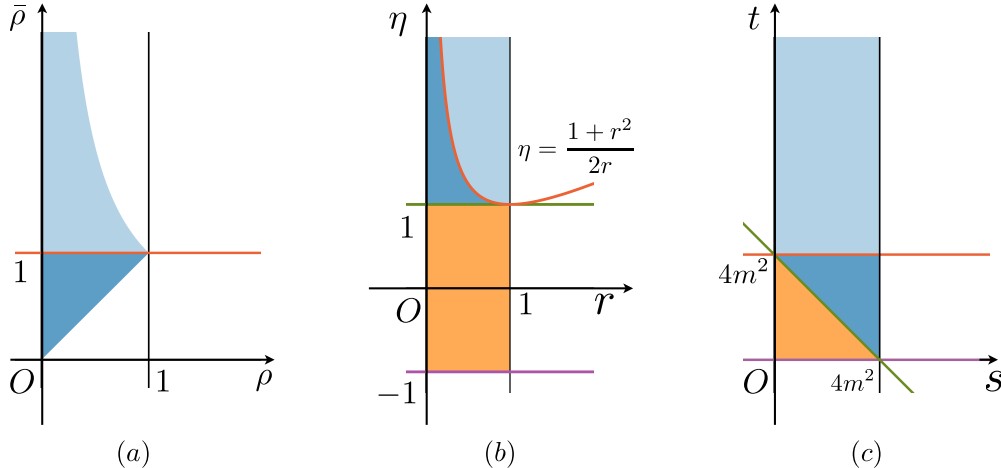

**Figure 1**. $\rho - \bar{\rho}$, $r - \eta$ and $s - t$ plane. Regions of the same colour are identified. The orange region corresponds to the Euclidean region where $z, \bar{z}$ (or $w, \bar{w}$) are complex conjugate of each other. The blue regions correspond to the Lorentzian region where $z, \bar{z}$ (or $w, \bar{w}$) are real and independent. This is also the integration region of the Lorentzian inversion formula (5.4). In the light blue region $\bar{z}$ (or $\bar{w}$) is continued around 1 and $1 < \bar{\rho} < 1/\rho$ (or $\bar{\rho} \to 1/\bar{\rho}$, $\rho \le \bar{\rho} \le 1$). The red lines and curves correspond to the branch points $\bar{\rho} = 1$, $1 + r^2 - 2r\eta = 0$ and $t = 4m^2$, respectively. The green and purple lines correspond to $\eta = 1, s + t = 4m^2$ and $\eta = -1, s = 0$, respectively.

Recovering the Froissart-Gribov formula from the Lorentzian inversion formula becomes perhaps most transparent using the conformal Mandelstam variables defined in equation (2.6). We see that the original integration domain is the region $0 < s < 4m^2$ with $4m^2 - s < t < 4m^2$, but taking the dDisc means that we also evaluate the correlator for all $t > 4m^2$.[7] For the original Froissart-Gribov formula we are instructed to integrate only over a semi-infinite line at fixed $s$ in the light blue region with $t > 4m^2$. We will soon see how this comes about through a partial saddle point analysis of the Lorentzian inversion formula.

**The integrand at large $\Delta$**

Let us now consider the behavior of the various ingredients in the Lorentzian inversion formula as all $\Delta$ become large. We will do so using the $(r, \eta)$ variables.

We will assume the validity of the amplitude conjecture, so we substitute $\mathcal{G}_{\mathrm{conn}}(r, \eta) \to \mathcal{G}_c(r, \eta) \mathcal{T}_R(r, \eta)$ and suppose that $\mathcal{T}_R(r, \eta)$ remains finite in the flat-space limit. The large $\Delta$ limit of the contact diagram was given in equation (3.25) which we reproduce here

$$\mathcal{G}_c(r, \eta) \xrightarrow{R \to \infty} \frac{w_c r^{-1/2}(r+1)^3}{\sqrt{(r^2 + 2r\eta + 1)(r^2 - 2r\eta + 1)}} \left(\frac{16r^2}{(r+1)^4}\right)^{\Delta_{\mathcal{O}}} \tag{5.7}$$

The first important realization is that the exponentially growing factor in the last parentheses is an increasing function of $r$. Therefore, keeping in mind figure 1(b), for each fixed $\eta$ the

---

[7]Sending $\bar{\rho} \to 1/\bar{\rho}$ corresponds to $(s, t, u) \to m^2(-4u, 16, -4s)/t$ which are the ($t$-channel version of the) 'tilded' variables of [21].

contribution of the secondary sheets (in light blue) will be exponentially larger than the contribution of the first sheet (in dark blue). We will therefore drop the latter.

On the secondary sheets we now furthermore observe that the factor $1 + r^2 - 2r\eta < 0$, so the square root in the above expression will give us a $\pm i$ depending on which sheet we are on. Taking this extra factor into account, and going back to the $(\rho, \bar\rho)$ variables, we may replace:

$$\text{dDisc}_t[\mathcal{G}_c(\rho, \bar\rho)\mathcal{T}_R(\rho, \bar\rho)] \rightarrow (-i)\,\mathcal{G}_c(\rho, 1/\bar\rho + i\epsilon)\frac{1}{2i}\left(\mathcal{T}_R(\rho, 1/\bar\rho + i\epsilon) - \mathcal{T}_R(\rho, 1/\bar\rho - i\epsilon)\right) \quad (5.8)$$

This is a nice result in itself: it shows how the three-term double discontinuity becomes the two-term single discontinuity (imaginary part) of the scattering amplitude.

The next step is to analyze the large $\Delta$ limit of the conformal block. Given that only the secondary-sheet contributions of the double discontinuity survive, it makes sense to first change variables from $\bar\rho$ to $\tilde\rho = 1/\bar\rho$ to get:

$$
\begin{aligned}
c_{\text{conn}}(\Delta, \ell) &= \kappa_{\Delta+\ell} \int_0^1 d\rho \int_1^{1/\rho} \frac{d\tilde\rho}{\tilde\rho^2}\, \mu(\rho, 1/\tilde\rho)G_{\ell+d-1,\Delta-d+1}(\rho, 1/\tilde\rho)\,\text{dDisc}_t\left[\mathcal{G}_{\text{conn}}(\rho, 1/\tilde\rho)\right] \\
&= \kappa_{\Delta+\ell} \int_0^1 d\rho \int_1^{1/\rho} \frac{d\tilde\rho}{\tilde\rho^2}\, \mu(\rho, 1/\tilde\rho)G_{\ell+d-1,\Delta-d+1}(\rho, 1/\tilde\rho)(-i)\mathcal{G}_c(\rho, \tilde\rho + i\epsilon)\text{Disc}_{\tilde\rho}\left[\mathcal{T}_R(\rho, \tilde\rho)\right] \\
&= \kappa_{\Delta+\ell} \int_0^1 dr \int_{\frac{1+r^2}{2r}}^\infty d\eta\, \hat\mu(r, \eta)G_{\ell+d-1,\Delta-d+1}\left(\rho = re^{i\phi}, \bar\rho = (re^{-i\phi})^{-1}\right) \\
&\qquad\qquad\qquad\qquad\qquad\qquad\qquad\qquad \times (-i)\mathcal{G}_c(r, \eta + i\epsilon)\text{Disc}_\eta\left[\mathcal{T}_R(r, \eta)\right], \quad (5.9)
\end{aligned}
$$

where in the last equality we set $\rho = re^{i\phi}$, $\tilde\rho = re^{-i\phi}$, and $\eta = \cos(\phi)$ as usual. The measure becomes

$$\hat\mu(r, \eta) = -2^{-d-1}\frac{(1-r^2)^{d-2}}{r^{d+1}}(\eta^2 - 1)^{\frac{d-3}{2}}(r^2 + 2r\eta + 1)(r^2 - 2r\eta + 1) \quad (5.10)$$

In appendix B we investigate what happens to the conformal block. The main result is[8]

$$
\begin{aligned}
\lim_{\Delta\to\infty} & g^{\text{pure}}_{\ell+d-1,\Delta-d+1}\left(\rho = re^{i\phi}, \bar\rho = (re^{-i\phi})^{-1}\right) \\
&= \frac{2^{d+2\ell+1}\Gamma\left(\frac{d}{2} + \ell\right)}{\sqrt{\pi}\Gamma\left(\frac{d-1}{2}\right)\Gamma(\ell+1)} \times \frac{r^{d-\Delta}Q_\ell^{(d)}(\eta)}{(1-r^2)^{\frac{d-2}{2}}\sqrt{-(r^2 + 2r\eta + 1)(r^2 - 2r\eta + 1)}},
\end{aligned}
$$
$$(5.11)$$

so here we recover in particular the Gegenbauer Q function. Note that the change of variables to $(r, \eta)$ is from the $(\rho, \tilde\rho)$ variables used in equation (5.9) rather than from the $(\rho, \bar\rho)$ variables.

**The saddle point**

We now see the pieces falling into place. A single conformal block splits into two 'pure'

---

[8]Here we have used the same normalization convention of conformal blocks as in [23].

blocks, but one of them is exponentially smaller and does not contribute to the saddle point. For the other one we find that the exponentially growing bits in the integrand are:

$$\int dr \ldots r^{-\Delta} \left( \frac{16r^2}{(r+1)^4} \right)^{\Delta_{\mathcal{O}}} \ldots \tag{5.12}$$

These bits are in fact exactly the same as in the Euclidean inversion formula discussed in the previous section, and they again localize the $r$ integral at the saddle point:

$$r_* = \frac{2\Delta_{\mathcal{O}} - \Delta}{2\Delta_{\mathcal{O}} + \Delta} = \frac{2m - \sqrt{s}}{2m + \sqrt{s}} \tag{5.13}$$

where for the last equality we used equation (3.32) to obtain the familiar relation between $r$ and the conformal Mandelstam variable given in equation (2.6). The integral over $\eta$ remains, and the prefactor becomes essentially the OPE density of the contact diagram $c_c(\Delta, 0)$. Altogether we can write:

$$\frac{c_{\text{conn}}(d/2 + R\sqrt{s}, \ell)}{c_c(d/2 + R\sqrt{s}, 0)} \xrightarrow{R \to \infty} n_\ell^{(d)} \frac{2\mathcal{N}_d}{\pi} \int_{t(\eta) = 4m^2}^{\infty} d\eta \, (\eta^2 - 1)^{-\frac{1}{2}} Q_\ell^{(d)}(\eta) \, \text{Disc}_t \left[ \mathcal{T}(s, t(\eta)) \right] , \tag{5.14}$$

which agrees exactly with equation (5.1). Note that the integration lower bound has been written in terms of $t$ using

$$t(\eta) = \frac{1}{2}(4m^2 - s)(1 + \eta) . \tag{5.15}$$

This is then how the Lorentzian inversion formula (5.4) can become the Froissart-Gribov formula (5.1) in the flat-space limit.

## 6 Analysis of the integration contour

The upshot of the previous sections was: if we assume the amplitude conjecture to hold, then the Euclidean and Lorentzian inversion formulas develop a saddle point whose contribution leads to the partial wave conjecture. The Euclidean inversion formula becomes the usual partial wave projection, whereas the Lorentzian inversion formula becomes the Froissart-Gribov formula.

As we mentioned above, the amplitude conjecture is in fact not always valid as there are blobs wherein the limit diverges. We discussed in subsection 4.1 that the $s$-channel blobs do not spoil the validity of our derivation, but the same cannot be said for the $t$- and $u$-channel blobs. But our derivation can *also* be invalidated in a different way. This can happen whenever the original integration contour, which is the interval $r \in (0, 1)$, cannot be deformed to the steepest descent contour through the saddle point. In this subsection we analyze the deformation of the integration contour in the general case, and we will show that it can lead

to issues with the partial wave conjecture even if the amplitude conjecture is assumed to be everywhere valid.

A convenient starting point for this discussion is as follows. Let us assume the amplitude conjecture is everywhere valid. Then we can imagine just performing the $\phi$ integral in the Euclidean inversion formula and the $\eta$ integral in the Lorentzian inversion formula exactly. Since the integrand is by assumption just the amplitude times an (associated) Legendre function, this by assumption just produces the partial wave $f_\ell(s(r))$, with the relation between $s$ and $r$ just the one given by the conformal Mandelstam variables of equation (2.6). All that is then left to do is the $r$-integral. For both inversion formulas this procedure then produces the following integral for $c(\Delta, \ell)$:

$$\int_0^1 dr\, r^{-3/2}(1-r)^{\frac{d}{2}-1}(1+r)^{\frac{d}{2}+2}r^{-\Delta}\left(\frac{r^2}{(1+r)^4}\right)^{\Delta_{\mathcal{O}}} f_\ell(s(r)) \tag{6.1}$$

and what we would like to check is whether this integral can be reliably evaluated via a saddle point approximation as $\Delta$ and $\Delta_{\mathcal{O}}$ become large so we obtain the amplitude conjecture. We recall that in both inversion formulas there is also a shadow term which is identical except for the replacement $\Delta \to d - \Delta$. This terms is subleading almost everywhere for $\text{Re}(\Delta) > d/2$, although we will see below that one cannot entirely ignore it.

**Integration endpoints**

Let us first analyze the endpoints of the integral. At $r = 0$ the integral converges only if $\text{Re}(\Delta) < 2\Delta_{\mathcal{O}}$ (for large $\Delta$ and $\Delta_{\mathcal{O}}$). This is partially an artefact of our approximations. As in the discussion of the $s$-channel exchange diagram, both inversion formulas can really only be trusted when $\Delta$ lives in the strip given by equation (4.3). Outside this strip the integral diverges near $r = 0$ which leads to a pole in $c(\Delta, \ell)$ at $\Delta_b$ and a corresponding non-analyticity in $f_\ell(s)$. We conclude that this endpoint is responsible for the 'right cut' in $f_\ell(s)$ which we therefore understand both physically and mathematically.

Let us now turn to the endpoint with $r = 1$. The contribution there can be estimated by expanding the integrand around $r = 1$, yielding

$$2^{-4\Delta_{\mathcal{O}}+2+\frac{d}{2}}\int dr(1-r)^{\frac{d}{2}-1}e^{-\Delta\log(1-r)}f_\ell(0) \approx \frac{2^{-4\Delta_{\mathcal{O}}+2+\frac{d}{2}}}{\Delta - d/2}f_\ell(0) \tag{6.2}$$

as the dominant contribution. Surprisingly, for some values of $\Delta$ (even inside the strip given in (4.3)) this term turns out to *dominate* over the contribution of the saddle point. This leads to a puzzle, because the exact expressions of simple cases like the contact and $s$-channel exchange diagram did not show any violation of the saddle point analysis. The puzzle is resolved by remembering the shadow term with $\Delta \to d - \Delta$: adding this term produce precisely the same contribution but with the opposite sign. It is this miracle which allows us to ignore any issues arising from the endpoint contribution near $r = 1$.

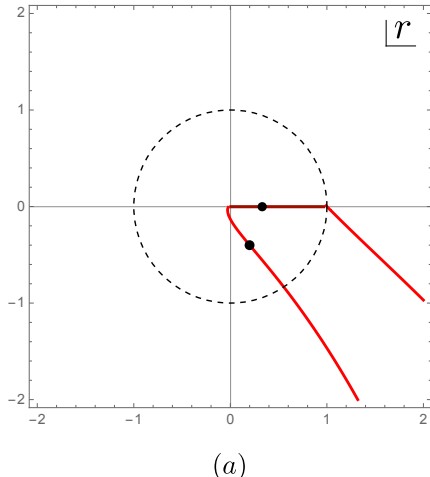
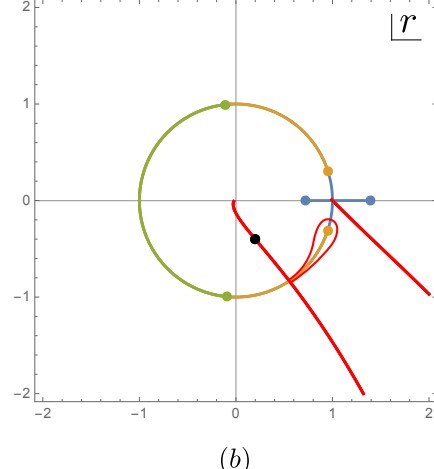

$(a)$                    $(b)$

**Figure 2**. $(a)$ Steepest descent contours and the corresponding saddle points in complex $r$-plane with $s = 1$ (dark red) and $\sqrt{s} = 1 + i$ (red). $(b)$ The same red steepest descent contour as in $(a)$ but with the presence of the branch cut in $f_\ell(s(r))$, with $t_0=3.9$ (blue), 4.1 (orange), 9 (green). The contour deformation to avoid the cut is illustrated for $t_0 = 4.1$.

### The steepest descent contour

In figure 2$(a)$ we plot, for two different values of $s$, (i) the saddle point, (ii) the steepest descent contour through the saddle point, and (iii) the steepest descent contour from the endpoint $r = 1$. We notice that there is no issue for $0 < s < 4m^2$, since for these real values of $s$ the steepest descent contour simply agrees with the original integration contour along the real axis. On the other hand, as soon as $s$ gets slightly complex the contour gets drastically deformed. Although one endpoint remains at $r = 0$, the other endpoint lies at infinity and so the contour necessarily exits the unit $r$ disk. We furthermore need to add another segment to return to the original endpoint at $r = 1$.

This new integration contour raises two potential issues. First, exiting the unit $r$ disk means that we are sending Mandelstam $s$ through the cut around 0. For fixed (real) angle $\phi$ this is actually not a region where we have much evidence for the general validity of the amplitude conjecture. In the next section we will discuss a specific example where the amplitude conjecture does not hold in that region, and show that it leads to additional divergences in the partial wave conjecture.

The second issue is due to the 'left cut' in $f_\ell(s)$, which is discussed in detail in the remainder of this section. We will be able to show, in full generality, that this limits the potential validity of the partial wave conjecture to a subset of the complex $s$ plane.

### The left cut in the partial waves

If the amplitude $T(s, t)$ has $t$-channel singularities, say a pole or branch cut starting at $t \geq t_0$, then the partial wave projection leads to a corresponding cut in the partial waves for

$s \leq 4m^2 - t_0$. This is the unavoidable left cut in the partial waves.[9]

Before we discuss the impact of this left cut, let us briefly consider the possible values of $t_0$. Non-perturbatively we expect $t_0$ to be at most $4m^2$, but for simple Feynman diagrams this is not necessarily the case. For example, in the $s$-channel exchange diagram there is no left cut whatsoever. On the other hand, in a $t$-channel exchange diagram the left cut in the partial waves starts at $t_0 = \mu^2$ with $\mu$ the mass of the exchanged particle.

In the $r$ variable the left cut is positioned as displayed in figure 2(b). Indeed, from equation (2.6) we find that the half-line $s \leq 0$ correspond to both the upper and lower half of the unit circle, with $r = -1$ corresponding to $s = -\infty$ and $r = 1$ corresponding to $s = 0$. We also find that the interval $0 < s < 4m^2$ corresponds to $1 > r > 0$. The endpoint of the cut gets mapped to:

$$r_{\text{cut}} = \frac{2m - \sqrt{4m^2 - t_0}}{2m + \sqrt{4m^2 - t_0}} \tag{6.3}$$

So if $t_0 = 4m^2$ then the cut is exactly along the entire unit circle, if $0 < t_0 < 4m^2$ then the cut in addition includes the real segment $0 < r_{\text{cut}} < r < 1$, and if $t_0 > 4m^2$ then the cut only spans the segment of the unit circle where $|\arg(r_{\text{cut}})| < |\arg(r)| < \pi$.

**The integration contour**

The integration contour must wrap around the left cut in the partial waves. It may therefore deviate from the steepest descent contour and this may spoil the validity of the partial wave conjecture. To see whether it does is a simple computation that starts from equation (6.1). One first needs to verify whether the steepest descent contour passes on the wrong side of $r_{\text{cut}}$ and, if so, whether this contour contribution dominates over the saddle point itself.

The result of this computation is shown for several values of $t_0$ in figure 3. (We have not considered $t_0 < 4m^2$ but will do so below.) We find a significant region in the complex $\Delta$ plane where the contour contribution to the integral (6.1) diverges and the partial wave conjecture is not valid. We stress that this holds for any partial wave $f_\ell(s)$ with a left cut and even if the amplitude conjecture is completely valid.

Notice that the divergent region encompasses the imaginary $\Delta/R$ axis above $\sqrt{|4m^2 - t_0|}$, and since $\Delta - d/2 = \sqrt{s}R$ this is precisely the region where the left cut should appear according to the partial wave conjecture. This answers the question of the appearance of the left cut in the partial wave conjecture in a somewhat disappointing way: the left cut is shrouded in a bigger region where the limit on the right-hand side simply does not exist.

## 7 The $t$-channel exchange diagram and discussions

In this section we will study the $t$-channel exchange diagram as our final example. It will allow us to concretize and extend the discussions of the left cut of the previous section.

---

[9]It is natural to think that the partial waves have only a left cut and a right cut, but this has not been proven from first principles. Here we will consider only the impact of the left cut on the domain of validity of the partial wave conjecture. Other non-analyticities can make this domain smaller.

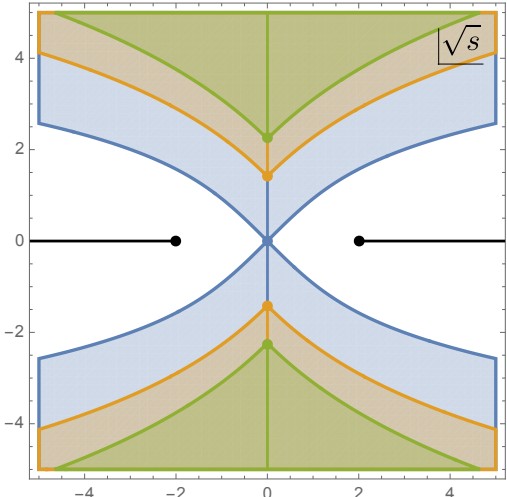

**Figure 3**. The divergence region on complex $\sqrt{s}$-plane with $t_0 = 4$ (blue), $6$ (orange), $9$ (green), where the dots indicate the location of the flat-space branch point $\sqrt{4 - t_0}$. The black dot and line are added by hand to indicate the flat-space two-particle threshold starting from $\sqrt{s} = 2$ and its shadow.

## 7.1 Flat-space expectations

In the flat-space limit we expect to recover the partial wave decomposition of the $t$-channel exchange flat-space Feynman diagram:

$$\mathcal{T}_\mu^{t\text{-exch}}(t) = \frac{-1}{t - \mu^2}, \tag{7.1}$$

where $\mu$ is the mass of the exchanged scalar particle and the coupling constant is set to one. Using the Froissart-Gribov formula, equation (5.1), it is a simple exercise to extract the partial wave coefficient. Indeed, the discontinuity of a simple pole (7.1) is just a delta function so the integral in (5.1) becomes trivial. The final result is then

$$f_{\ell,\mu}^{t\text{-exch}}(s) = \frac{2\mathcal{N}_d}{4m^2 - s} \left( \left( 1 - \frac{2\mu^2}{4m^2 - s} \right)^2 - 1 \right)^{\frac{d-3}{2}} Q_\ell^{(d)} \left( -1 + \frac{2\mu^2}{4m^2 - s} \right), \tag{7.2}$$

which is actually valid for all spins $\ell$ because equation (7.1) vanishes fast enough at large $|t|$.

We note that (7.2) is analytic in the entire $s$ plane except for the possible branch cut in the prefactor as well as in the $Q_\ell^{(d)}(\cdot)$, which both occur when

$$\left( -1 + \frac{2\mu^2}{4m^2 - s} \right)^{-2} \geq 1 \qquad \Longleftrightarrow \qquad s \leq 4m^2 - \mu^2. \tag{7.3}$$

So in this perturbative example we can dial $\mu^2$ to move the starting point of the left cut in the partial wave along the real axis.

## 7.2 Blobs on the second sheet

Above we reviewed the result from [15] that the flat-space limit of the Witten exchange diagram diverges in certain blobs $D_\mu$ in the complex plane of the corresponding Mandelstam invariant. The flat-space results (7.1) is then only recovered outside of these blobs. We also mentioned that these blobs are empty whenever $\mu > 2m$, but in actuality this is only true on the first sheet. In the previous section we have seen that the steepest descent contour in the partial wave conjecture exits the unit $r$ disk, *i.e.* it extends to secondary sheets of the conformal correlator. Our first order of business is therefore to extend the position-space analysis to these regions.

Concretely, we ask what happens in say an $s$-channel scalar exchange diagram on the sheet where we rotate $s$ around $4m^2$. This corresponds to rotating $r$ around 0 through the cut in the negative real axis. The Mellin space representation [39] is no longer valid in this region, so the flat-space derivation in the appendix of [12] cannot be used. Instead we have to use the method of [15] which directly used the saddle-point approximation of the position-space integral involving bulk-boundary and bulk-bulk propagators. As discussed in [15], in this picture the divergences in the blobs for $\mu < 2m$ simply correspond to the contribution of an extra pole which gets picked up when moving the integration contour to the steepest descent contour. This analysis is easily extended to the second sheet, for which the result is shown in figure 4.

In figure $4(a)$ we first recall the first-sheet analysis from [15]. We find that an additional pole only gets picked up in the lightly shaded regions, but generally its contribution is subleading. Only in the darkly shaded regions does the extra contribution dominate and in fact diverge. These are the regions $D_\mu$. They are non-empty only for $\mu < 2m$ and then their leftmost point coincides with the pole at $s = \mu^2$.

As we send $s$ around $4m^2$ we find an entirely different picture on the second sheet. Here the pole is always picked up. For $\mu < 2$ this first of all leads to the same divergent region as on the first sheet, but in addition there is trouble in a much bigger region to the left of it. For $\mu > 2$ we likewise find a big domain where the flat-space limit diverges.

## 7.3 Impact on the partial waves

In the previous section we noticed that the left cut in the flat-space partial waves $f_\ell(s)$ led automatically to an issue in the partial wave conjecture. In doing so we assumed that the amplitude conjecture was everywhere valid, including on a secondary sheet. But the analysis of the previous subsection shows that this is in fact *not* the case for the exchange diagram: even for $\mu > 2m$ there is a big region on the second sheet where the amplitude conjecture fails.

The implication for the partial wave conjecture is now as follows. Consider equation (6.1). The integrand here features $f_\ell(s)$, obtained by integrating the position-space amplitude against the Legendre polynomial, and in the preceding discussion we assumed that it was everywhere finite, even for $r$ outside the unit disk which corresponds to rotating $s$ around

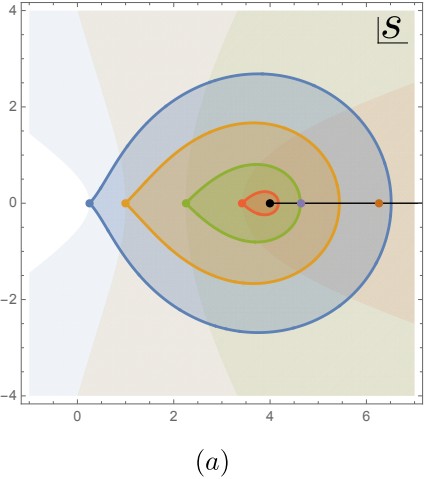
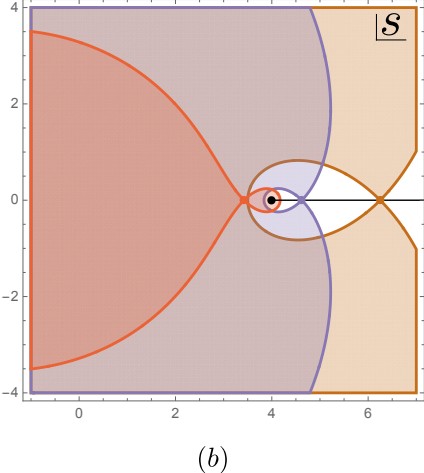

$$(a) \qquad\qquad\qquad (b)$$

**Figure 4**. The divergence region of $s$-channel exchange Witten diagram: $(a)$ on the first sheet with $\mu=0.5$ (blue), 1, 1.5, 1.85, 2.15, 2.5 (brown) (the pole leading to the divergence is not picked up outside the light shaded region for $\mu < 2$ or anywhere for $\mu > 2$); $(b)$ on the second sheet with $\mu=1.85$ (red), 2.15, 2.5 (brown). The coloured dots indicate the flat-space poles and they lie on the tip/cross point of the divergence regions. The black dot and the half line indicate the threshold at $s = 4$ and the branch cut attached to it.

0. This assumption is now violated for the $t$-channel exchange diagram. To see this we can just consider the standard partial wave projection, where $f_\ell(s)$ is obtained by integrating the amplitude over $t$ from $t = 0$ to $t = 4m^2 - s$. If we rotate $s$ around 0 then the endpoint of the $t$ integral rotates around $4m^2$, which brings us right into the bad region on the second sheet.

Since the main contribution comes from the endpoint at $t = 4m^2 - s$ we can immediately transform the bad region in the $t$ plane to the $s$ plane and then the $r$ plane. This leads to figure 5. Inside the bad regions the integrand in (6.1) is not $f_\ell(s)$ but rather diverges exponentially. Of course at the same time there is an exponential suppression from the rest of the integrand (if we move along the steepest descent contour, since the saddle point is always inside the unit disk). So to see whether a divergence actually occurs requires a bit of computation.

## 7.4 New saddle points

Let us now analyze the inversion of the $t$-channel exchange diagram without assuming the amplitude conjecture. Using conformal Mandelstam variables we can write

$$\mathcal{G}_{\text{t-exch}}(s,t) = \mathcal{G}_c(s,t) \left( \frac{-1}{t - \mu^2} + E_\mu^{(t)}(s,t) \right), \qquad (7.4)$$

where we have denoted the divergence blob as the "error" term $E_\mu^{(t)}(s,t)$. As discussed in section 7.2, $E_\mu^{(t)}$ exists everywhere (but does not always dominate) on the second sheet but this is not the case on the first sheet (see figure 4).

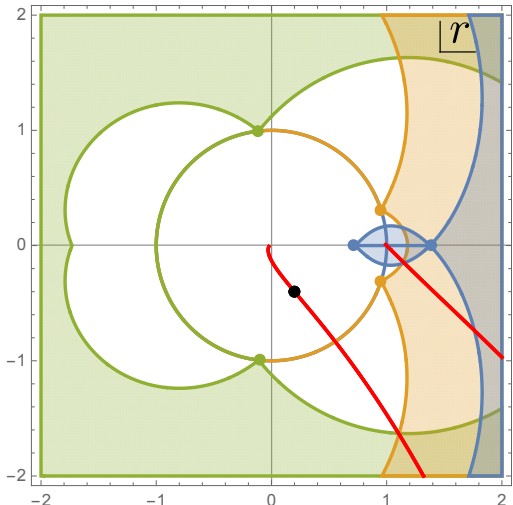

**Figure 5**. The steepest descent contour in complex $r$-plane with the presence of the second-sheet divergent region of the $t$-channel exchange Witten diagram, with $s = \sqrt{1+i}$, $t_0$=3.9 (blue), 4.1 (orange), 9 (green). When $t_0 < 4$ the blob also appears on the first sheet (inside the unit disk of $r$-plane). To be compared with figure 2.

The inversion of the first term in (7.4) follows the previous analysis in Section 6, whereas the second term leads to new contributions. For the inversion of $E_\mu(s,t)$, the $t$ integral localises at the endpoint $t = 4m^2 - s$. Plugging this into (6.1) we have an extra contribution of the form

$$\int_0^1 dr\, r^{-3/2}(1-r)^{\frac{d}{2}-1}(1+r)^{\frac{d}{2}+2}r^{-\Delta}\left(\frac{r^2}{(1+r)^4}\right)^{\Delta_\mathcal{O}} E_\mu^{(t)}(s, 4m^2 - s)\,, \tag{7.5}$$

up to some unimportant factors. The leading large exponent can be found using

$$\frac{1}{R}\log E_\mu^{(t)}(s, 4m^2 - s) = (2m + \mu)\log\left(\frac{2m+\mu}{2m+\sqrt{4m^2-s}}\right) + (2m - \mu)\log\left(\frac{2m-\mu}{2m-\sqrt{4m^2-s}}\right)\,. \tag{7.6}$$

Combining this with the other exponent factors (recall $r = r(s)$ from equation (2.6)), we find saddle points for equation (7.5) at:

$$s_\pm = \frac{4(x^2 - \mu^2) + \mu^2(x^2 + \mu^2) \pm 4\mu\sqrt{x^2(\mu^2 + x^2 - 4)}}{(x^2 + \mu^2)^2}\,, \qquad x \equiv \frac{\Delta}{R}\,. \tag{7.7}$$

It turns out that the $s$ integral localises to the $s_-$ saddle point. By comparing the integral at this new saddle point with the previous saddle point (5.13), we find the region of divergence in the complex plane of $\sqrt{s}$, i.e. of $(\Delta - d/2)/R$, as shown in figure 6.

Note that the bad region shown in figure 3 is a strict subset of the bad region shown in

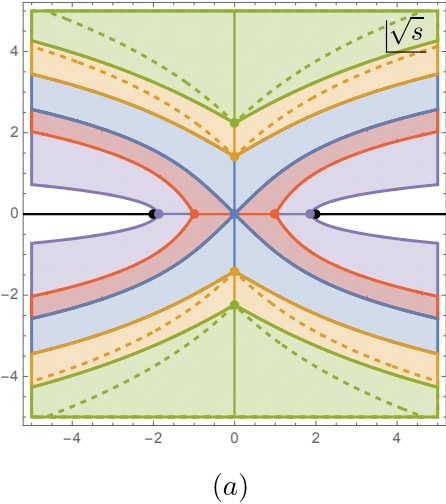
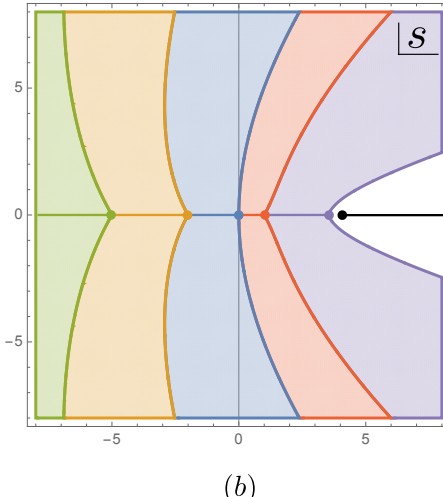

$(a)$ $(b)$

**Figure 6**. The full bad regions for $t$-exchange on $(a)$ complex $\sqrt{s} = (\Delta - d/2)/R$ plane and $(b)$ complex $s$-plane with $t_0=0.5$ (purple), 3, 4, 6, 9 (green). The dots indicate the flat-space left branch point $s = 4 - t_0$ and they lie on the boundary of the bad regions. The regions with dashed boundaries are the same as those in figure 3.

figure 6. The former was a non-perturbative analysis that followed entirely from the existence of a left cut of $f_\ell(s)$ and thus provides a universal restriction on the domain of validity of the partial wave conjecture. The partial wave conjecture for the $t$-channel exchange diagram is not valid in a slightly larger domain due to the divergences in position space on the second sheet.

## 7.5 Numerical check in two dimensions

In this subsection we perform a numerical check of the above observations and results, in particular of figure 6. This is possible because the OPE density of the $t$-channel exchange Witten diagram can be computed exactly in two and four dimensions using the Lorentzian inversion formula (see e.g. [40, 41]). We will first briefly review how the computation is done in two dimensions, and then evaluate the obtained OPE density numerically in the flat-space limit.

### 7.5.1 OPE density in two dimensions

The $t$-channel exchange Witten diagram has the following conformal block decomposition (see for example [42])

$$W^{t\text{-exch}}_{\hat{\Delta},0}(P_i) = \frac{1}{(P_{23}P_{14})^{\Delta_{\mathcal{O}}}} \left( A_{\hat{\Delta},\Delta_{\mathcal{O}}} \, G_{\hat{\Delta},0}(1-z,1-\bar{z}) + \sum_{n=0}^{\infty} B_{n,0} G_{2\Delta_{\mathcal{O}}+2n,0}(1-z,1-\bar{z}) \right.$$
$$\left. + \sum_{n=0}^{\infty} C_{n,0}\partial_n G_{2\Delta_{\mathcal{O}}+2n,0}(1-z,1-\bar{z}) \right) ,$$
(7.8)

with[10]

$$A_{\hat{\Delta},\Delta_{\mathcal{O}}} = R^{5-d} \frac{\Gamma\left(\frac{\hat{\Delta}}{2}\right)^4 \Gamma\left(\frac{1}{2}(2\Delta_{\mathcal{O}}-\hat{\Delta})\right)^2 \Gamma\left(\frac{1}{2}(-d+\hat{\Delta}+2\Delta_{\mathcal{O}})\right)^2}{128\pi^{\frac{3d}{2}}\Gamma(\hat{\Delta})\Gamma\left(-\frac{d}{2}+\hat{\Delta}+1\right)\Gamma\left(-\frac{d}{2}+\Delta_{\mathcal{O}}+1\right)^4} ,$$
(7.9)

where we set all the external dimensions equal to $\Delta_{\mathcal{O}}$ and denoted the dimension of the exchanged operator by $\hat{\Delta}$,

The double discontinuity of the diagram only gets a contribution from the first 'single-trace' conformal block in (7.8). To compute the corresponding OPE density $c_{t\text{-exch}}(\Delta,\ell)$ one therefore needs to integrate this single-trace block against the 'inverted' conformal block appearing in the Lorentzian inversion formula (5.9). This can be done explicitly in $d=2$, where the blocks factorize and the integrals reduce to

$$\Omega_{h,h',p} \equiv \int_0^1 \frac{dz}{z^2} \left(\frac{z}{1-z}\right)^p k_{2h}(z)k_{2h'}(1-z) .$$
(7.10)

which can be written in closed form as [40, 41]

$$\Omega_{h,h',p} = \frac{\Gamma(2h)\Gamma(h'-p+1)^2\Gamma(h-h'+p-1)}{\Gamma(h)^2\Gamma(h+h'-p+1)} \, {}_4F_3\left(\begin{array}{c} h',h',h'-p+1,h'-p+1 \\ 2h',h'+h-p+1,h'-h-p+2 \end{array}\right)$$
$$+ \frac{\Gamma(2h')\Gamma(h'-h-p+1)\Gamma(h+p-1)^2}{\Gamma(h')^2\Gamma(h'+h+p-1)} \, {}_4F_3\left(\begin{array}{c} h+p-1,h+p-1,h,h \\ h'+h+p-1,2h,-h'+h+p \end{array}\right) .$$
(7.11)

Altogether we find that, again for $d=2$,

$$c_{t\text{-exch}}(\Delta,\ell) = 2\kappa_{\Delta+\ell}A_{\hat{\Delta},\hat{\ell};\Delta_{\mathcal{O}}} \sin^2\left(\pi\frac{\hat{\Delta}-\hat{\ell}-2\Delta_{\mathcal{O}}}{2}\right) \Omega_{1-h,\frac{\hat{\Delta}}{2},\Delta_{\mathcal{O}}}\Omega_{\bar{h},\frac{\hat{\Delta}}{2},\Delta_{\mathcal{O}}} ,$$
(7.12)

---

[10]To align with the normalisation given in footnote 5 we need to multiply $A_{\hat{\Delta},\Delta_{\mathcal{O}}}$ by $\mathcal{C}_{\Delta}^{-2}$.

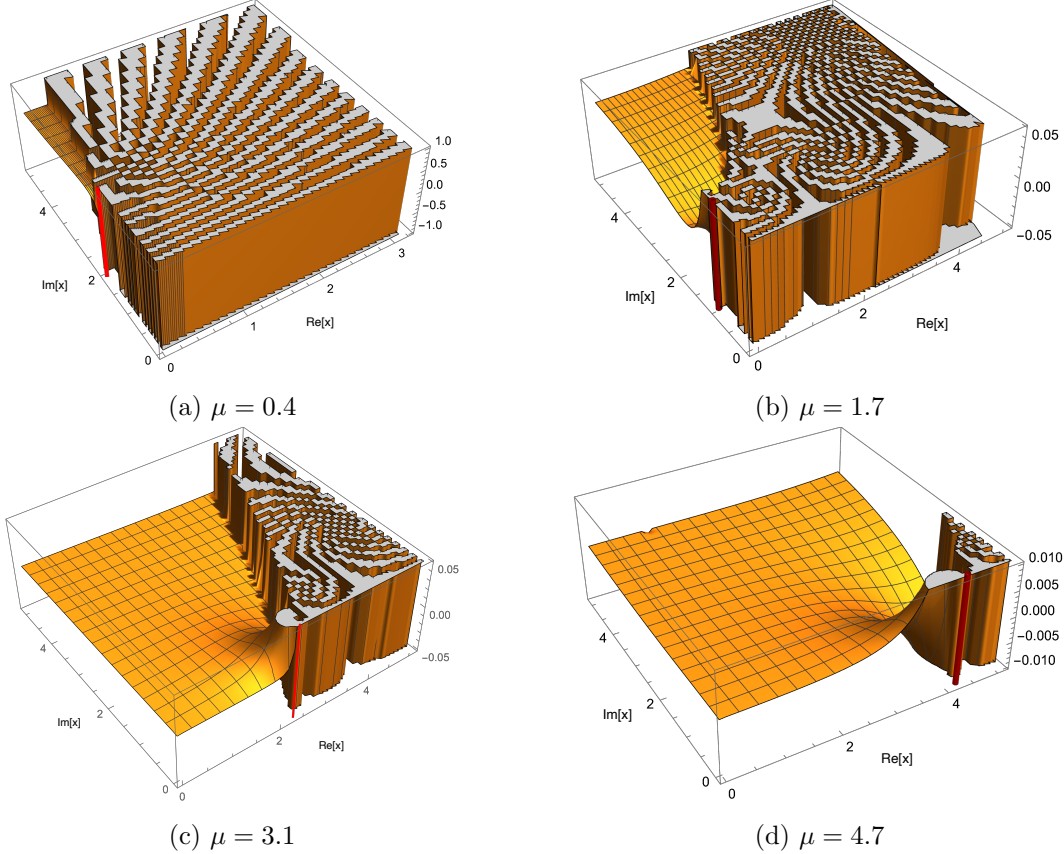

(a) $\mu = 0.4$            (b) $\mu = 1.7$

(c) $\mu = 3.1$            (d) $\mu = 4.7$

**Figure 7**. Real part of $\frac{c_{t\text{-exch}}(\Delta,\ell)}{c_{\mathrm{c}}(\Delta,0)}$ (which is expected to become the partial wave coefficient in the large $R$ limit through (3.35)) in $2d$ with $m = 1, \ell = 2, R = 80.1$ and various $\mu$ on the complex plane of $x$ ($\Delta = d/2 + iRx$). The highly oscillating regions indicate a clear breakdown of (3.35). The red lines indicate the location of the flat-space branch points (7.14) and they are located within the oscillating region. Only the top right quadrant is plotted.

Our aim is now to see if this density, when plugged into the right-hand side of the partial wave conjecture (1.2), reproduces the flat-space answer (7.2) outside of the bad regions of figure 6 and diverges inside of them.

To fully exploit the factorisation of $z, \bar{z}$ integral in $2d$, we also use saddle point approximation of (7.10) to calculate the flat-space limit of (7.12). The analysis turns out to be very involved and the details are given in Appendix D, but the results completely match the intuition gathered in the previous sections.

### 7.5.2 Numerical test in two dimensions

We can do a numerical check at large but finite $R$. For our plots we will use the variable $x$ defined through

$$\Delta = \frac{d}{2} + R\sqrt{s} = \frac{d}{2} + iRx \tag{7.13}$$

In particular, the branch point (7.3) appears now at

$$x^2 = \mu^2 - 4m^2 \,. \tag{7.14}$$

We will only analyse the top right quadrant of the complex $x$ plane because the other quadrants are related by shadow symmetry and real-analyticity of the OPE density.

In figure 7 we plot the right-hand side of the partial wave conjecture at finite $\Delta_{\mathcal{O}} = mR = 80.1$ and for various values of $\mu = \hat{\Delta}/\Delta_{\mathcal{O}}$. We have also chosen to plot only the real part of the OPE density for $\ell = 2$, but the plots look entirely similar for imaginary part as well as for the other spins.

The main takeaway is entirely in agreement with the saddle-point analyses of the previous subsection. For each $\mu$ there is an obvious 'bad' region where the flat-space limit diverges and oscillates. In those regions the partial wave conjecture clearly breaks down since the limit does not exist. These divergences precisely mask the appearance of the left cut, the beginning of which is marked with the red line in the figure.

We also explicitly verified that for the same values of $t_0 = \mu^2$, the boundary of the divergence regions from analytics and numerics coincide (albeit not exactly because $R$ is not infinite in numerics). In particular, we also find divergences in the numerical result if we choose $\sqrt{s}$ in the small gap between the dashed region and the solid region in figure 6.

## 8 Connection with other partial wave prescriptions

### 8.1 The flat-space limit of the principal series decomposition

In subsection 3 we showed that the saddle point contribution in the Euclidean inversion formula, equation (3.14), reduces to the partial wave projection wherever the amplitude conjecture is valid. Let us now consider the fate of its inverse, equation (3.11), which expresses the correlation function in terms of the OPE density $c(\Delta, \ell)$. We call it the principal series decomposition.

The first step is an easy exercise. In the region where the partial wave conjecture holds the connected OPE density $c_{\mathrm{conn}}(\Delta, \ell)$ by assumption reduces to $c_c(\Delta, 0)$ times a finite function $n_\ell^{(d)} f_\ell(s)$ in the flat-space limit. Substituting also the large $\Delta$ behavior of the conformal block, we find that (the second line of) equation (3.11) develops a saddle point which localizes the $\Delta$ integral at the familiar location $\Delta = d/2 + R\sqrt{s}$. The sum over spins remains, and altogether the saddle point contribution reproduces precisely the standard partial wave decomposition of the amplitude as given in equation (3.2).

The next step is to analyze the steepest descent contour. It is drawn for two values of $s$ in figure 8. We see in particular that it passes close to the natural branch point at $\Delta = 2\Delta_\phi$, corresponding to $s = 4m^2$, whenever $s$ is chosen close to a physical value. From this integration contour we can understand the following things.

First, its location is compatible with the existence of the $s$-channel blobs in position space: if there are poles in the OPE density corresponding to blocks with a dimension below

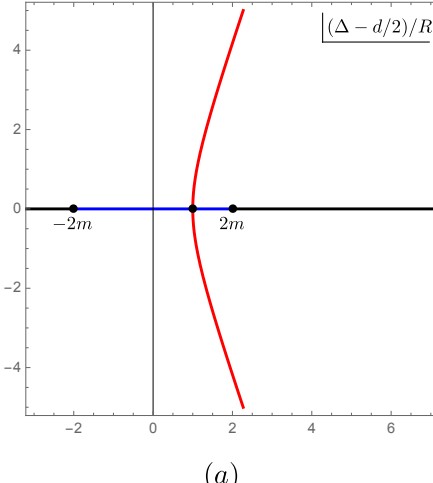 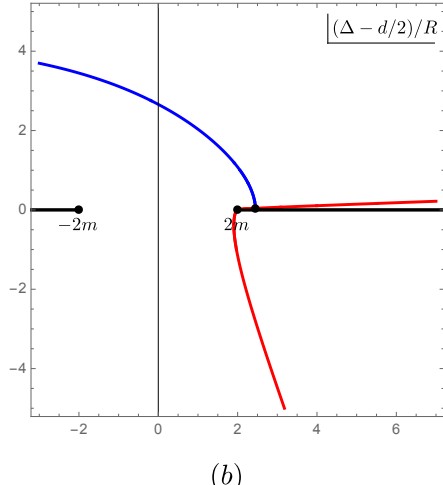

(a)                                      (b)

**Figure 8**. The steepest descent contours (red) for (3.11) with (a) $s = 1$ and (b) $s = 6 + 0.2i$. In blue we also show the steepest ascent contour.

$2\Delta_\phi$ then the original integration contour and the steepest descent contour lie on opposite sides of these poles. The additional contribution from the poles leads to potential divergences which leads to the position-space blobs in the $s$-channel.

Second, there are two types of additional contributions which we can now argue must generally be subleading since otherwise they would invalidate the consistency with our derivations of sections 3 and 5. The first type are those from the bad regions in $f_\ell(s)$ shown in figure 6. These are apparently supressed by the exponential falloff along the steepest descent contour, since they are nowhere to be found in the position-space expression wherever the amplitude conjecture holds. The second type of additional contributions is due to the spurious poles in the OPE density, which are needed to offset poles in the conformal blocks themselves (see figure 9 for visualisation and [23, 43, 44] for discussions). But the OPE density for the contact diagram happens to not have such spurious poles, and since we also do not expect them to appear in $f_\ell(s)$ they must disappear altogether. So we claim that the residues of the spurious poles should also vanish in the flat-space limit for the partial wave conjecture to make sense.

Finally we note that this inverse analysis provides some (indirect) evidence for the validity of the partial wave conjecture outside the strip with $d - \Delta_b < \mathrm{Re}(\Delta) < \Delta_b$. This extended domain of validity cannot be deduced from the Euclidean and Lorentzian inversion formulas since, as we discussed around equation (4.3), they stop converging at the boundary of the strip. It is of course also in agreement with the various examples discussed above.

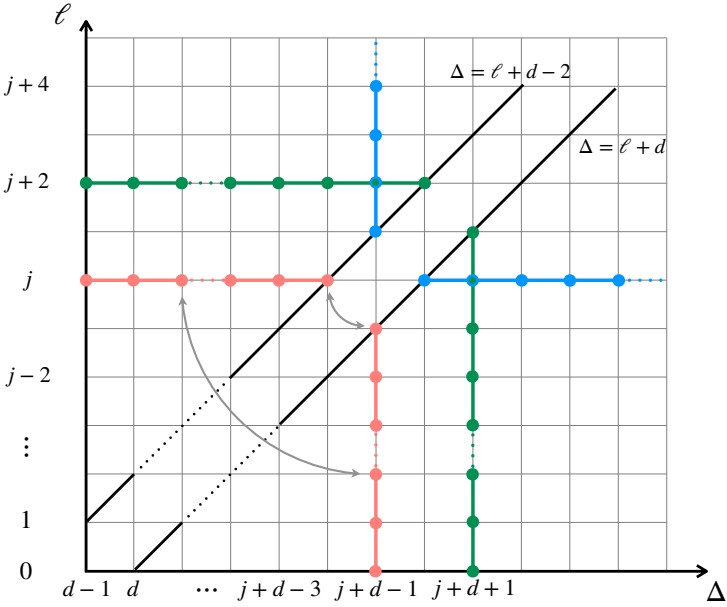

**Figure 9**. Cancellation of spurious poles in the principal series decomposition (3.11). In the upper left we find the poles in the conformal block at $\Delta = \ell + d - 1 - k$ with residue proportional to $G_{\ell+d-1,\ell-k}$, $k = 1, 2, \ldots, \ell$ [35, 36]. In the bottom right are the spurious poles in the OPE density which coincide with the locations of the residue blocks. In a general correlator the poles with the same colour cancel against each other.

## 8.2 Connection with hyperbolic partial waves

In [21] we introduced the *trimmed* amplitude $T^{\text{trim}}(s, t, u)$, which was defined as:

$$\mathcal{G}_{\text{gff}}(s, \eta) + \mathcal{G}_{\text{c}}(s, \eta) T^{\text{trim}}(s, \eta) = 1 + \sum_{k \text{ with } \Delta_k \geq 2\Delta_{\mathcal{O}}} a_k^2 G_{\Delta_k, \ell_k}(s, \eta) . \tag{8.1}$$

Here we used the conformal Mandelstam variables $(s, \eta)$ which are related to the conformal cross ratios (or rather $(z, \bar{z})$) through equation (2.6). The removal of the conformal blocks with $\Delta < 2\Delta_\phi$ certainly destroys many of the nice properties of the correlation function, but for physical $s$ we argued in [21] that their effect is simply to remove the $s$-channel blobs (also see appendix C). Therefore $T^{\text{trim}}(s, t, u)$ should actually become the scattering amplitude in the flat-space limit for $s > 4m^2$ and $t, u < 0$. We then perform the standard partial wave projection to define:

$$f_\ell^{\text{trim}}(s) := \frac{\mathcal{N}_d}{2} \int_{-1}^1 d\eta (1 - \eta^2)^{\frac{d-3}{2}} P_\ell^{(d)}(\eta) T^{\text{trim}}(s, \eta) . \tag{8.2}$$

These *trimmed partial waves* should reduce to the ordinary partial waves for physical kinematics. (They are equivalent to the connected part of the hyperbolic partial waves defined in [21].)

How can we relate the trimmed partial waves of equation (8.2) to the partial wave conjecture? The trimmed correlator is defined by removing all the conformal blocks with $\Delta < 2\Delta_\phi$, and one may naively think that it can only be described with an entirely different OPE density, something like $c^{\text{trim}}(\Delta, \ell)$. But we can in fact also describe it with the original OPE density $c(\Delta, \ell)$ if we just use an integration contour such that only the poles with $\Delta > 2\Delta_\phi$ get picked up. If we call such a contour $\gamma^{\text{trim}}$ then we can write, for the connected part,

$$T^{\text{trim}}(s, \eta) = \sum_{l=0}^{\infty} \int_{\gamma^{\text{trim}}} d\Delta \, c_{\text{conn}}(\Delta, \ell) \frac{G_{\Delta, \ell}(s, \eta)}{\mathcal{G}_{\text{c}}(s, \eta)} \,. \tag{8.3}$$

We stress that the choice of contour is the only difference between this equation and the standard split decomposition (as in the second line of (3.11)) for the connected correlator. Next we substitute this decomposition in equation (8.2), take the large $\Delta$ limit of the conformal block as given in equation (3.27) and of the contact diagram as given in equation (3.25). We then see that the $\eta$ integral can be done exactly and kills the sum over $\ell$. This leaves us with the $\Delta$ integral. The by now familiar saddle point analysis then yields:

$$n_\ell^{(d)} f_\ell^{\text{trim}}(s) = \lim_{R \to \infty} \left. \frac{c_{\text{conn}}(\Delta, \ell)}{c_c(\Delta, 0)} \right|_{\Delta = \frac{d}{2} + R\sqrt{s}} \tag{8.4}$$

which ties the partial wave conjecture of this paper to the hyperbolic partial waves of [21]. The above derivation should work for $s$ nearly physical. Its added value is that there is manifestly no additional contribution that invalidates the saddle point because there are no divergent contour contributions. Indeed, unlike the original integration contour, the contour $\gamma^{\text{trim}}$ can actually be deformed to the steepest descent contour shown in figure 8(b) without passing through additional singularities. This is how the trimmed correlators also naturally appear also in our partial wave conjecture.

We can also make contact with the phase shift formula of [12] where the relation $\Delta \sim R\sqrt{s}$ first appeared. This formula expresses the partial waves in terms of the OPE data and should work for real $s$. It can be related to the amplitude conjecture through a saddle point in the conformal block decomposition [15]. In the same way it can also be immediately connected to the hyperbolic partial wave and therefore also to the partial wave conjecture by the above reasoning. These latter two therefore offer the natural extension to complex $s$ of the phase shift formula.

## 9 Flat-space limit of the conformal dispersion relation

In this section we show how the two-variable conformal dispersion relation [24, 45] becomes the single-variable flat-space dispersion relation in the flat-space limit, assuming the amplitude conjecture (2.3) remains valid throughout the integration domain.

## 9.1 Derivation assuming the amplitude conjecture

We will focus on the connected part of the four-point correlator. If we split it into two parts

$$\mathcal{G}_{\text{conn}}(z, \bar{z}) = \mathcal{G}_{\text{conn}}^t(z, \bar{z}) + \mathcal{G}_{\text{conn}}^u(z, \bar{z}), \tag{9.1}$$

then the fixed-$s$ dispersion relation in [24] is

$$\mathcal{G}_{\text{conn}}^t(z, \bar{z}) = \int_0^1 dw d\bar{w} K^s(z, \bar{z}, w, \bar{w}) \, \text{dDisc}_t[\mathcal{G}_{\text{conn}}(w, \bar{w})], \tag{9.2}$$

where the integration kernel is

$$K^s(z, \bar{z}, w, \bar{w}) = K_B^s \theta \left( \rho_z \bar{\rho}_z \bar{\rho}_w - \rho_w \right) + K_C^s \frac{d\rho_w}{dw} \delta \left( \rho_w - \rho_z \bar{\rho}_z \bar{\rho}_w \right). \tag{9.3}$$

We recall the relation

$$z = \frac{4\rho_z}{1 + \rho_z^2} \tag{9.4}$$

and similarly for $\bar{\rho}_z$ as well as $\rho_w$ and $\bar{\rho}_w$.

When all external dimensions are equal the integration kernel has the following explicit form [24]

$$K_C^s = \frac{4}{\pi} \frac{1}{\bar{w}^2} \left( \frac{1 - \rho_z^2 \bar{\rho}_z^2 \bar{\rho}_w^2}{(1 - \rho_z^2)(1 - \bar{\rho}_z^2)(1 - \bar{\rho}_w^2)} \right)^{1/2} \frac{1 - \rho_z \bar{\rho}_z \bar{\rho}_w^2}{(1 - \rho_z \bar{\rho}_w)(1 - \bar{\rho}_z \bar{\rho}_w)}, \tag{9.5}$$

$$K_B^s = -\frac{1}{64\pi} \left( \frac{z\bar{z}}{w\bar{w}} \right)^{3/2} \frac{(\bar{w} - w) \left( \frac{1}{w} + \frac{1}{\bar{w}} + \frac{1}{z} + \frac{1}{\bar{z}} - 2 \right)}{((1-z)(1-\bar{z})(1-w)(1-\bar{w}))^{\frac{3}{4}}} x^{\frac{3}{2}} \, {}_2F_1 \left( \frac{1}{2}, \frac{3}{2}, 2, 1-x \right), \tag{9.6}$$

$$x \equiv \frac{\rho_z \bar{\rho}_z \rho_w \bar{\rho}_w \left( 1 - \rho_z^2 \right) \left( 1 - \bar{\rho}_z^2 \right) \left( 1 - \rho_w^2 \right) \left( 1 - \bar{\rho}_w^2 \right)}{(\bar{\rho}_z \bar{\rho}_w - \rho_w \rho_z)(\rho_z \bar{\rho}_w - \rho_w \bar{\rho}_z)(\rho_z \bar{\rho}_z - \rho_w \bar{\rho}_w)(1 - \rho_w \rho_z \bar{\rho}_w \bar{\rho}_z)}. \tag{9.7}$$

We will show that these equation reduce to a fixed-$s$ dispersion relation for the flat-space scattering amplitude:

$$\mathcal{T}(s, t(\eta)) = \frac{1}{\pi} \int_{\eta_0}^\infty \frac{d\eta'}{\eta' - \eta} \, \text{Disc}_{t'} \left[ \mathcal{T} \left( s, \eta' \right) \right] + (t \leftrightarrow u), \tag{9.8}$$

where $\eta_0$ is the branch point. More precisely, assuming (2.3) we will show that

$$\int_0^1 dw d\bar{w} K^s(z, \bar{z}, w, \bar{w}) \frac{\text{dDisc}_t[\mathcal{G}(w, \bar{w})]}{\mathcal{G}_c(z, \bar{z})} \xrightarrow{R \to \infty} \frac{1}{\pi} \int_{\eta_0}^\infty \frac{d\eta'}{\eta' - \eta} \, \text{Disc}_{t'} \left[ \mathcal{T} \left( s, \eta' \right) \right], \tag{9.9}$$

where the identifications between $z, \bar{z}$ and $s, \eta'$ are given in (2.4), (2.5) and (2.6).

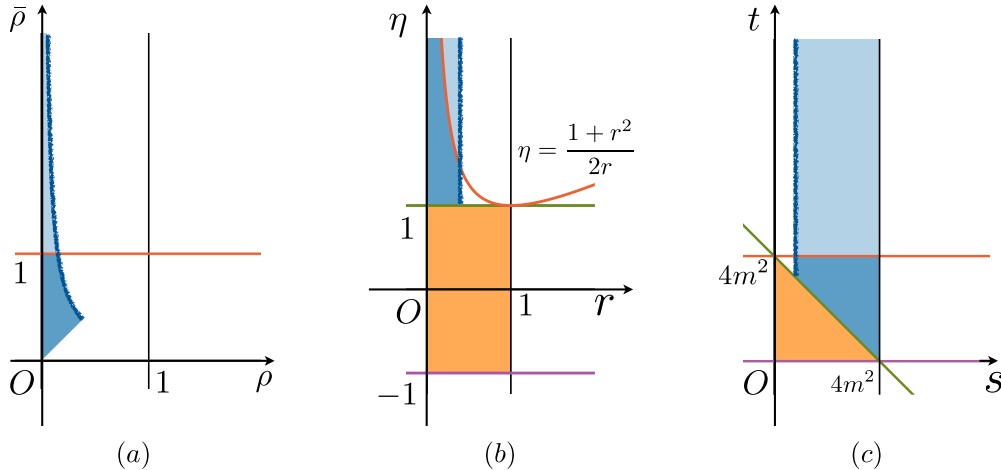

**Figure 10**. The integration region of the conformal dispersion relation on the $\rho - \bar{\rho}$, $r - \eta$ and $s - t$ planes. The dark blue fuzzy line corresponds to $s_w = s_z$ which is the support of the delta function in equation (9.3).

### 9.1.1 The integration domain

In figure 10 we show the effective integration domain for the conformal dispersion relation in various coordinates. The theta function in (9.3) selects a subset of the full integration domain of the Lorentzian inversion formula which we showed in figure 1. The delta function adds an extra contribution on the boundary of this subset.

The integration domain looks especially nice when we transform to the conformal Mandelstam variables defined in equation (2.6). We see that the delta function is simply supported at $s_w = s_z$, and the integration domain simply extends to the right of it, see figure 10(c).

The expectation that is naturally deduced from the figure is that (9.9) comes about from the $K_C$ term only, with the $K_B$ contribution vanishing in the flat-space limit. This is indeed what happens as we will now proceed to show.

### 9.1.2 $K_C$ part

Let us denote

$$\mathcal{G}^t_{\text{conn}}(z, \bar{z}) = \mathcal{G}^{t, K_B}_{\text{conn}}(z, \bar{z}) + \mathcal{G}^{t, K_C}_{\text{conn}}(z, \bar{z})$$

$$\mathcal{G}^{t, K_B}_{\text{conn}}(z, \bar{z}) = \int_0^1 dw d\bar{w} K_B^s \theta(\rho_z \bar{\rho}_z \bar{\rho}_w - \rho_w) \, d\mathrm{Disc}_t[\mathcal{G}_{\text{conn}}(w, \bar{w})] \,,$$

$$\mathcal{G}^{t, K_C}_{\text{conn}}(z, \bar{z}) = \int_0^1 dw d\bar{w} K_C^s \frac{d\rho_w}{dw} \delta(\rho_w - \rho_z \bar{\rho}_z \bar{\rho}_w) \, d\mathrm{Disc}_t[\mathcal{G}_{\text{conn}}(w, \bar{w})] \,,$$

(9.10)

according to the split in (9.3) and first focus on the $K_C$ part. Changing integration variable to $\rho_w, \bar{\rho}_w$ we have

$$\frac{\mathcal{G}^{t,K_C}_{\text{conn}}(\rho_z, \bar{\rho}_z)}{\mathcal{G}_c(\rho_z, \bar{\rho}_z)} = \int_0^1 d\bar{\rho}_w K_C^s \frac{d\bar{w}}{d\bar{\rho}_w} \frac{\mathrm{dDisc}_t[\mathcal{G}_{\text{conn}}(\bar{\rho}_w \rho_z \bar{\rho}_z, \bar{\rho}_w)]}{\mathcal{G}_c(\rho_z, \bar{\rho}_z)}, \qquad (9.11)$$

where the delta function in the $K_C$ part has immediately removed one of the integrals for us and set $\rho_w = \bar{\rho}_w \rho_z \bar{\rho}_z$. The remaining integral over $\bar{\rho}_w$ will later be identified with the $\eta'$-integral on the RHS of (9.9).

Using the flat-space limit of the double discontinuity (5.8) we get

$$\frac{\mathcal{G}^{t,K_C}_{\text{conn}}(\rho_z, \bar{\rho}_z)}{\mathcal{G}_c(\rho_z, \bar{\rho}_z)} \xrightarrow{R \to \infty} \int_0^1 d\bar{\rho}_w K_C^s \frac{d\bar{w}}{d\bar{\rho}_w} \frac{\mathcal{G}_c(\bar{\rho}_w \rho_z \bar{\rho}_z, 1/\bar{\rho}_w + i\epsilon)}{i\mathcal{G}_c(\rho_z, \bar{\rho}_z)} \mathrm{Disc}_{t'}[\mathcal{T}(s, t'(\eta_w))]. \qquad (9.12)$$

Recall that in the flat-space limit, the contribution to the double discontinuity from the principal-sheet correlator is subdominant and omitted. This leads to the identification

$$r_w = \sqrt{\frac{\rho_w}{\bar{\rho}_w}} = \sqrt{\rho_z \bar{\rho}_z} = r_z, \qquad (9.13)$$

where the first equality only holds when $\bar{w}$ is continued around 1, and the second equality follows from the delta function $\delta(\rho_z \bar{\rho}_z \bar{\rho}_w - \rho_w)$. This identification is important because from (3.25) we know that the contact diagram's exponential dependence on $\Delta_\mathcal{O}$ is completely determined by $r$, thus we immediately see that the exponential growth from the two contact diagrams in equation (9.12) cancel with each other. Using

$$\bar{\rho}_w = \frac{1}{r_w \left( \eta_w + \sqrt{\eta_w^2 - 1} \right)} \qquad (9.14)$$

we find[11]

$$\frac{\mathcal{G}_c(\bar{\rho}_w \rho_z \bar{\rho}_z, 1/\bar{\rho}_w + i\epsilon)}{i\mathcal{G}_c(\rho_z, \bar{\rho}_z)} = \sqrt{\frac{(1 + r_z^2)^2 - 4r_z^2 \eta_z^2}{4r_z^2 \eta_w^2 - (1 + r_z^2)^2}}. \qquad (9.15)$$

We now change integration variable to $\eta_w$ and rewrite the integral as

$$\frac{\mathcal{G}^{t,K_C}_{\text{conn}}(\rho_z, \bar{\rho}_z)}{\mathcal{G}_c(\rho_z, \bar{\rho}_z)} \xrightarrow{R \to \infty} \int_{\frac{1+r_z^2}{2r_z}}^{\infty} d\eta_w K_C^s \frac{d\bar{w}}{d\eta_w} \sqrt{\frac{(1 + r_z^2)^2 - 4r_z^2 \eta_z^2}{4r_z^2 \eta_w^2 - (1 + r_z^2)^2}} \mathrm{Disc}_{t'}[\mathcal{T}(s, t'(\eta_w))]$$

$$= \frac{1}{\pi} \int_{\frac{1+r_z^2}{2r_z}}^{\infty} d\eta_w \frac{1}{\eta_w - \eta_z} \mathrm{Disc}_{t'}[\mathcal{T}(s, t'(\eta_w))], \qquad (9.16)$$

---

[11] The relative minus sign between the numerator and the denominator is because the $i$ factor from analytically continuing $\mathcal{G}_c(w, \bar{w})$ around $\bar{w} = 1$ has been absorbed into $\mathrm{Disc}_{t'}[\mathcal{T}(s, t')]$.

where to get the final result we have used (9.5) and

$$\frac{d\bar{w}}{d\eta_w} = \frac{4r_z\left(\sqrt{\eta_w^2-1}-\eta_w\right)\left(r_z-\eta_w+\sqrt{\eta_w^2-1}\right)}{\sqrt{\eta_w^2-1}\left(r_z+\eta_w-\sqrt{\eta_w^2-1}\right)^3}\,. \tag{9.17}$$

Therefore, the $K_C$ part reduces to the flat-space dispersion relation, as expected.

### 9.1.3 $K_B$ part

Next let us turn to the $K_B$ part. Resolving the theta function's constraint we have

$$\frac{\mathcal{G}_{\text{conn}}^{t,K_C}(\rho_z,\bar{\rho}_z)}{\mathcal{G}_c(\rho_z,\bar{\rho}_z)} = \int_0^1 d\bar{\rho}_w \int_0^{r_z^2\bar{\rho}_w} d\rho_w\,\frac{dw}{\rho_w}\frac{d\bar{w}}{\bar{\rho}_w}K_B\frac{\mathrm{dDisc}_t[\mathcal{G}_{\text{conn}}(w,\bar{w})]}{\mathcal{G}_c(\rho_z,\bar{\rho}_z)}\,. \tag{9.18}$$

Then applying (5.8) we get

$$\frac{\mathcal{G}_{\text{conn}}^{t,K_B}(\rho_z,\bar{\rho}_z)}{\mathcal{G}_c(\rho_z,\bar{\rho}_z)} \xrightarrow{R\to\infty} \int_0^1 d\bar{\rho}_w \int_0^{r_z^2\bar{\rho}_w} d\rho_w\,\frac{dw}{d\rho_w}\frac{d\bar{w}}{d\bar{\rho}_w}K_B\frac{\mathcal{G}_c(\rho_w,1/\bar{\rho}_w+i\epsilon)}{i\mathcal{G}_c(\rho_z,\bar{\rho}_z)}\,\mathrm{Disc}_{\bar{\rho}_w}[\mathcal{T}(\rho_w,1/\bar{\rho}_w)]\,. \tag{9.19}$$

The $\rho_w$ integral localizes because of the large exponential factor

$$-i\mathcal{G}_c(\rho_w,1/\bar{\rho}_w+i\epsilon) \sim \left(\frac{16\rho_w\bar{\rho}_w}{(\rho_w+\bar{\rho}_w+2\sqrt{\rho_w\bar{\rho}_w})^2}\right)^{\Delta_{\mathcal{O}}} =: e^{\Delta_{\mathcal{O}}f(\rho_w)}\,. \tag{9.20}$$

The function $f(\rho_w)$ has a maximum at

$$\rho_w^* = \bar{\rho}_w\,, \tag{9.21}$$

and the second-order derivative is

$$f''(\rho_w^*) = -\frac{1}{4\bar{\rho}_w^2} < 0\,, \tag{9.22}$$

so the steepest descent contour through this saddle point coincides with the real-$\rho_w$ axis. However, the theta function restricts the range of $\rho_w$ to

$$\rho_w \le \rho_z\bar{\rho}_z\bar{\rho}_w = r_z^2\bar{\rho}_w \le \rho_w^*\,, \quad r_z \in [0,1]\,. \tag{9.23}$$

which means that the saddle point is never reached. Instead, $f(\rho_w)$ is maximized at the integration endpoint

$$\rho_w = r_z^2\bar{\rho}_w\,. \tag{9.24}$$

This is of course precisely (9.13) which corresponds to the support locus of the delta function

in the $K_C$ part. This time, however, the integral over $\rho_w$ brings in an additional factor of $\Delta_{\mathcal{O}}^{-1}$ (or $\Delta_{\mathcal{O}}^{-1/2}$ if the saddle point coincides with the endpoint). This key factor makes the $K_B$ part smaller than the $K_C$ part and negligible in the flat-space limit.

### 9.1.4 Discussion

In contrast to the flat-space limits of the Euclidean and Lorentzian inversion formulas, the above derivation goes through without any deformations of the integration contour. We thus recover a dispersion relation as long as (a) the amplitude conjecture is obeyed everywhere along the integration domain, and (b) the integration kernel does not blow up at infinity. In practice, however, point (a) is never really obeyed and then a more refined analysis is needed to really prove the existence of a dispersion relation in the flat-space limit. For example, in [21] this was done by using the Polyakov-Regge block decomposition, but it would be interesting to see whether other derivations exist.

As for point (b), we note that our derivation immediately extends to subtracted conformal dispersion relations like those introduced in [24, 45]. These subtrated conformal dispersion kernels are just the original one multiplied by rational functions of the cross ratios, thus this extra factor simply goes along the ride in the flat-space limit. This means that any polynomial divergence can actually be brought under control.[12]

## 10 Conclusions

The partial wave conjecture (1.2) follows naturally from a saddle point approximation to both the Euclidean and the Lorentzian inversion formulas. A more detailed analysis of the steepest descent contour however shows that it necessarily fails whenever $f_\ell(s)$ has a left cut, in the domains shown in figure 3. The actual domain of divergence can however be even larger if the amplitude conjecture fails on the secondary sheets, as shown in figure 6 for the $t$-channel exchange diagram. Finally we showed that the conformal dispersion relation for conformal correlators reduces to a single-variable dispersion relation for amplitudes whenever the amplitude conjecture is valid.

In Table 1 we have summarised various relations among $(d+1)$-dimensional QFT objects and $d$-dimensional boundary conformal theory ($\mathrm{BCT}_d$) objects built up through the flat-space limit of QFT in $\mathrm{AdS}_{d+1}$. Many of these results are completely in line with expectations, and indeed a very similar table appeared in Section 9 of [34]. For gapped theories there is however a precise way in which the *analogies* between these structures become *equalities* upon taking the flat-space limit. This is the main point the table is meant to convey.

Of course, once one passes to well-defined equations there is also a greater risk that things go wrong. We have indeed repeatedly seen that the flat-space limit necessarily diverges

---

[12]In conformal subtracted dispersion relations without taking the flat-space limit, the subtraction points should not matter, but when $R \to \infty$, it is safest to set the subtraction points to coincide with the branch points at $w, \bar{w} = 0, 1$, as was done in [24, 45]. This guarantees that no extra poles are picked up during contour deformation, which could lead to extra divergent blobs in the flat-space limit.

| QFT$_{d+1}$ | BCT$_d$ |
|---|---|
| $\langle \underline{\tilde{k}}_1 \ldots \underline{\tilde{k}}_a \| S \| \underline{k}_1 \ldots \underline{k}_b \rangle$ | $\langle \mathcal{O}(\tilde{n}_1) \ldots \mathcal{O}(\tilde{n}_a) \mathcal{O}(n_1) \ldots \mathcal{O}(n_b) \rangle\|_{\text{S-matrix}}$ (1.1) |
| $T(s,t,u)$ | $\mathcal{G}_{\text{conn}}(s,t,u)/\mathcal{G}_c(s,t,u)$ (2.3) |
| $\text{Disc}\, T(s,t,u)$ | $\text{dDisc}\, \mathcal{G}_{\text{conn}}(s,t,u)/\mathcal{G}_c(s,t,u)$ (5.8) |
| Dispersion relations (9.8) | CFT Dispersion relations (9.2) |
| $f_\ell(s)$ | $c_{\text{conn}}(\Delta,\ell)/c_c(\Delta,0)\big\|_{\Delta=d/2+R\sqrt{s}}$ (3.35) |
| Partial wave projection (3.6) | Euclidean inversion formula (3.14) |
| Froissart-Gribov formula (5.1) | Lorentzian inversion formula (5.4) |
| Unitarity condition (3.8) | CFT unitarity (C.13) |

**Table 1**. The list of $(d + 1)$-dimensional QFT structures that can be found from $d$-dimensional conformal objects through the flat-space limit of a gapped QFT in AdS$_{d+1}$. Conformal Mandelstam variables have been adopted for the right column.

for certain kinematics. In all cases so far this can be traced back to divergent contour contributions. These can sometimes be interpreted physically via AdS Landau diagram, and in other times are logically unavoidable such as when the partial waves have a left cut. This paper provides a further step in deducing where these equalities do and do not hold.

## Acknowledgements

We thank Miguel Correia, Shota Komatsu, João Penedones, Jiaxin Qiao and Sasha Zhiboedov for useful discussions and suggestions. We are also grateful to the co-organizers and participants of the "Bootstrap 2023" at ICTP-SAIFR, São Paulo and the "S-matrix bootstrap V" at SwissMAP Research Station, les Diablerets for providing a stimulating environment where part of this work was completed. The authors are supported by Simons Foundation grant #488649 (XZ) and #488659 (BvR and XZ) for the Simons Collaboration on the non-perturbative bootstrap. XZ is also supported by the Swiss National Science Foundation through the project 200020 197160 and through the National Centre of Competence in Research SwissMAP. BvR is also funded by the European Union (ERC, QFTinAdS, 101087025). Views and opinions expressed are however those of the author(s) only and do not necessarily reflect those of the European Union or the European Research Council Executive Agency. Neither the European Union nor the granting authority can be held responsible for them.

# A  Principal series representation of Witten diagrams

In this appendix we review the general way to obtain the principal series representation for four-point Witten diagrams whose bulk part can be written as a two-point function $F(X_1, X_2)$. For more details see [14, 46–49].

We start from[13]

$$\mathcal{G}(P_i) = \prod_{i=1}^{4} \mathcal{C}_{\Delta_i}^{-\frac{1}{2}} \int dX_1 dX_2 G_{B\partial}^{\Delta_1}(X_1, P_1) G_{B\partial}^{\Delta_2}(X_1, P_2) F(X_1, X_2) G_{B\partial}^{\Delta_3}(X_2, P_3) G_{B\partial}^{\Delta_4}(X_2, P_4). \tag{A.1}$$

Using the spectral representation for $SO(d+1, 1)$ invariant two-point functions in AdS

$$F(X_1, X_2) = \int_{-\infty}^{\infty} \frac{d\nu}{2\pi} \widehat{F}(\nu) \Omega_\nu(X_1, X_2), \tag{A.2}$$

$$\Omega_\nu(X_1, X_2) = 2\nu^2 R^{d-1} \int_{\partial \text{AdS}} dQ \, G_{B\partial}^{\frac{d}{2}+i\nu}(X_1, Q) G_{B\partial}^{\frac{d}{2}-i\nu}(X_2, Q), \tag{A.3}$$

$$G_{B\partial}^{\Delta}(X, P) = \frac{\mathcal{C}_\Delta}{R^{(d-1)/2}(-2P \cdot X/R)^\Delta}, \qquad \mathcal{C}_\Delta = \frac{\Gamma(\Delta)}{2\pi^{\frac{d}{2}} \Gamma(\Delta - \frac{d}{2} + 1)}, \tag{A.4}$$

we find

$$\prod_{i=1}^{4} \mathcal{C}_{\Delta_i}^{\frac{1}{2}} \mathcal{G}(P_i) = \int dX_1 dX_2 G_{B\partial}^{\Delta_1}(X_1, P_1) G_{B\partial}^{\Delta_2}(X_1, P_2) G_{B\partial}^{\Delta_3}(X_2, P_3) G_{B\partial}^{\Delta_4}(X_2, P_4)$$

$$\times \int_{-\infty}^{\infty} \frac{d\nu}{2\pi} \widehat{F}(\nu) 2\nu^2 R^{d-1} \int_{\partial \text{AdS}} dQ \, G_{B\partial}^{\frac{d}{2}+i\nu}(X_1, Q) G_{B\partial}^{\frac{d}{2}-i\nu}(X_2, Q) \tag{A.5}$$

---

[13]The prefactor is included to align with the normalisation in footnote 5.

Integrating over $X_1, X_2$ and $Q$ we get[14]

$$\prod_{i=1}^{4} \mathcal{C}_{\Delta_i}^{\frac{1}{2}} \mathcal{G}(P_i) = \int_{-\infty}^{\infty} \frac{d\nu}{2\pi} 2\nu^2 R^{3d+1} \widehat{F}(\nu) f_{\Delta_1 \Delta_2 (\frac{d}{2}+i\nu)} f_{\Delta_3 \Delta_4 (\frac{d}{2}-i\nu)}$$

$$\times \int_{\partial AdS} dQ \langle \mathcal{O}_1(P_1) \mathcal{O}_2(P_2) \mathcal{O}_{\frac{d}{2}+i\nu}(Q) \rangle \langle \widetilde{\mathcal{O}}_{\frac{d}{2}-i\nu}(Q) \mathcal{O}_3(P_3) \mathcal{O}_4(P_4) \rangle$$

$$= T^{\Delta_i}(P_i) \int_{-\infty}^{\infty} \frac{d\nu}{2\pi} 2\nu^2 R^4 \widehat{F}(\nu) f_{\Delta_1 \Delta_2 (\frac{d}{2}+i\nu)} f_{\Delta_3 \Delta_4 (\frac{d}{2}-i\nu)} \Psi_{\frac{d}{2}+i\nu,0}^{\Delta_i}(z,\bar{z}),$$

$$= T^{\Delta_i}(P_i) \int_{-\infty}^{\infty} \frac{d\nu}{2\pi} 4\nu^2 R^4 \widehat{F}(\nu) f_{\Delta_1 \Delta_2 (\frac{d}{2}+i\nu)} f_{\Delta_3 \Delta_4 (\frac{d}{2}-i\nu)} K_{\frac{d}{2}-i\nu,0}^{\Delta_{34}} G_{\frac{d}{2}+i\nu,0}(z,\bar{z}),$$

$$(A.6)$$

where [49, 50]

$$f_{\Delta_1 \Delta_2 \Delta_3} = \frac{\pi^{\frac{d}{2}}}{2} \frac{\mathcal{C}_{\Delta_1} \mathcal{C}_{\Delta_2} \mathcal{C}_{\Delta_3}}{\Gamma(\Delta_1) \Gamma(\Delta_2) \Gamma(\Delta_3)} \Gamma(\Delta_{123}/2) \Gamma(\Delta_{12\tilde{3}}/2) \Gamma(\Delta_{231}/2) \Gamma(\Delta_{312}/2), \qquad (A.7)$$

with $\Delta_{ijk} \equiv \Delta_i + \Delta_j - \Delta_k$. From the above expression we may read off the desired OPE density:

$$c(\Delta = \frac{d}{2} + i\nu, \ell = 0) = 4\nu^2 R^4 \widehat{F}(\nu) f_{\Delta_1 \Delta_2 (\frac{d}{2}+i\nu)} f_{\Delta_3 \Delta_4 (\frac{d}{2}-i\nu)} K_{\frac{d}{2}-i\nu,0}^{\Delta_{34}} \qquad (A.8)$$

in terms of $\widehat{F}(\nu)$.

Let us give two examples. First, for the contact diagram [49]

$$F_c(X_1, X_2) = \delta^{(d+1)}(X_1, X_2), \qquad \widehat{F}_c(\nu) = R^{-d-1}, \qquad (A.9)$$

and for $s$-channel scalar exchange diagram [46]

$$F_{s\text{-exch}}(X_1, X_2) = G_{BB}(X_1, X_2), \qquad \widehat{F}_{s\text{-exch}}(\nu) = \frac{R^{-d+1}}{(\Delta_b - d/2)^2 + \nu^2}. \qquad (A.10)$$

Using the technique in [49], the above results can be generalised to contact diagrams from vertices with arbitrary derivatives and to exchange diagrams with spinning propagators.

---

[14]Note that for each integral over a bulk point we get a factor of $R^{d+1}$

$$\int dX_1 G_{B\partial}^{\Delta_1}(X_1, P_1) G_{B\partial}^{\Delta_2}(X_1, P_2) G_{B\partial}^{\frac{d}{2}+i\nu}(X_1, Q) = R^{d+1} f_{\Delta_1 \Delta_2 (\frac{d}{2}+i\nu)} \langle \mathcal{O}_1(P_1) \mathcal{O}_2(P_2) \mathcal{O}_{\frac{d}{2}+i\nu}(Q) \rangle.$$

## A.1 Recovering the position-space expression

As a check of our OPE density, let us recover the large $\Delta$ expression of the contact Witten diagram given in [15]. Using (A.9) we can start from

$$\prod_{i=1}^{4}\mathcal{C}_{\Delta_i}^{\frac{1}{2}}\mathcal{G}_c(P_i)=T^{\Delta_i}(P_i)\int_{-\infty}^{\infty}\frac{d\nu}{2\pi}\frac{4\nu^2}{R^{d-3}}f_{\Delta_1\Delta_2(\frac{d}{2}+i\nu)}f_{\Delta_3\Delta_4(\frac{d}{2}-i\nu)}K_{\frac{d}{2}-i\nu,0}^{\Delta_{34}}G_{\frac{d}{2}+i\nu,0}(z,\bar{z}). \quad (A.11)$$

In the flat-space limit we then have[15]

$$\prod_{i=1}^{4}\mathcal{C}_{\Delta_i}^{\frac{1}{2}}\mathcal{G}_c(P_i)\xrightarrow{R\to\infty}\frac{1}{(P_{12})^{\Delta_{\mathcal{O}}}(P_{34})^{\Delta_{\mathcal{O}}}}\int_{-\infty}^{\infty}\frac{d\nu}{2\pi}\frac{4\nu^2}{R^{d-3}}f_{\Delta_{\mathcal{O}}\Delta_{\mathcal{O}}(\frac{d}{2}+i\nu)}f_{\Delta_{\mathcal{O}}\Delta_{\mathcal{O}}(\frac{d}{2}-i\nu)}K_{\frac{d}{2}-i\nu,0}N(r,\eta)(4r)^{\frac{d}{2}+i\nu}$$

$$=\frac{R^{3-d}}{(P_{12})^{\Delta_{\mathcal{O}}}(P_{34})^{\Delta_{\mathcal{O}}}}\int_{-\infty}^{\infty}\frac{d\nu}{2\pi}f(\rho,\bar{\rho})\frac{(i\nu)^{\frac{d}{2}-1}}{\left(\Delta_{\mathcal{O}}^2+\frac{\nu^2}{4}\right)^{\frac{d}{2}+1}}e^{g(\nu)}$$

$$(A.12)$$

where

$$f(\rho,\bar{\rho})=\frac{\Delta_{\mathcal{O}}^{2d-2}}{8\pi^{\frac{3d}{2}-1}}\frac{\left(\sqrt{\rho}\sqrt{\bar{\rho}}\right)^{\frac{d}{2}}}{\sqrt{(1-\rho^2)\,(1-\bar{\rho}^2)}(1-\rho\bar{\rho})^{\frac{d}{2}-1}} \quad (A.13)$$

and

$$g(\nu)=2\left(\Delta_{\mathcal{O}}+\frac{i\nu}{2}\right)\log\left(\Delta_{\mathcal{O}}+\frac{i\nu}{2}\right)+2\left(\Delta_{\mathcal{O}}-\frac{i\nu}{2}\right)\log\left(\Delta_{\mathcal{O}}-\frac{i\nu}{2}\right)$$
$$-4\Delta_{\mathcal{O}}\log(\Delta_{\mathcal{O}})+\frac{i\nu}{2}\log\rho\bar{\rho}. \quad (A.14)$$

Using the saddle point approximation, we find that the saddle point sits at

$$\nu^*=-2imR\frac{1-\sqrt{\rho\bar{\rho}}}{1+\sqrt{\rho\bar{\rho}}}=-iR\sqrt{s}, \quad (A.15)$$

and the one-loop fluctuation factor is

$$g''(\nu^*)=\frac{(1+\sqrt{\rho\bar{\rho}})^2}{4\Delta_{\mathcal{O}}\sqrt{\rho\bar{\rho}}}=\frac{4m^2}{\Delta(4m^2-s)}, \quad (A.16)$$

---

[15]Here we used the block normalisation (3.28).

so we ultimately find the limit

$$\mathcal{G}_c\left(P_i\right) \overset{R\to\infty}{\longrightarrow} \frac{1}{(P_{12})^{\Delta_\mathcal{O}}(P_{34})^{\Delta_\mathcal{O}}} \frac{1}{2\pi i} \sqrt{\frac{2\pi}{Re^{-i\pi}g''(\nu^*)}} (4r)^{\frac{d}{2}+i\nu^*} c(\nu^*,0)N(r,\eta,0)$$

$$= w_c 2^{2\Delta_\mathcal{O}-\frac{1}{2}} \frac{\left(\sqrt{P_{12}P_{34}} + \sqrt{P_{13}P_{24}} + \sqrt{P_{14}P_{23}}\right)^{-2\Delta_\mathcal{O}+3/2}}{(P_{12}P_{13}P_{14}P_{23}P_{24}P_{34})^{1/4}}, \tag{A.17}$$

with

$$w_c = 2^{-\frac{d+7}{2}} \pi^{-\frac{d-1}{2}} \Delta_\mathcal{O}^{\frac{d-5}{2}} R^{3-d}, \tag{A.18}$$

which (taking into account the different normalization) exactly matches with the result in [15].

# B Large spin limit of pure power block in general spacetime dimension

In this appendix we calculate the large spin limit of the "pure block" defined in [23] in general space time dimension. This is useful for taking the flat-space limit of the Lorentzian inversion formula in Section 5. The large spin limit of the pure block has also been worked out in [51].

The pure block is defined through splitting the conformal block into two parts:

$$\mathcal{G}_{\ell+d-1,\Delta+1-d}(z,\bar{z}) = (\mathcal{G}^{\text{pure}})_{\ell+d-1,\Delta+1-d}(z,\bar{z})$$
$$+ 2^{d-2\Delta} \frac{\Gamma(\Delta-1)\Gamma\left(-\Delta+\frac{d}{2}\right)}{\Gamma\left(\Delta-\frac{d}{2}\right)\Gamma(-(\Delta+1-d))} (\mathcal{G}^{\text{pure}})_{\ell+d-1,-\Delta+1}(z,\bar{z}), \tag{B.1}$$

where each of the $\mathcal{G}^{\text{pure}}$ can be expanded into pure power terms in the limit $z \ll \bar{z} \ll 1$ [23]. For the first term we have[16]

$$(G^{\text{pure}})_{\ell+d-1,\Delta-d+1}(z,\bar{z}) = z^{d-1-\frac{\Delta-\ell}{2}} \bar{z}^{\frac{\Delta+\ell}{2}}, \qquad z \ll \bar{z} \ll 1. \tag{B.2}$$

In the large spin ($\Delta \to \infty$ in this case) the pure block becomes

$$\lim_{\Delta\to\infty} (G^{\text{pure}})_{\ell+d-1,\Delta-d+1}\left(\rho = r(\eta^2 - \sqrt{\eta^2-1}), \bar{\rho} = r^{-1}(\eta^2 + \sqrt{\eta^2-1})^{-1}\right) = \alpha_\ell^{(d)} \, H_\ell^{(d)}(r,\eta)$$

$$H_\ell^{(d)}(r,\eta) \equiv \frac{r^{d-\Delta}}{(1-r^2)^{\frac{d-2}{2}} \sqrt{4r^2\eta^2 - (1+r^2)^2}} Q_\ell^{(d)}(\eta), \tag{B.3}$$

where $\alpha_\ell^{(d)}$ is a normalization factor to be fixed. This can be checked by using the quadratic

---

[16] Here we have used the same normalization convention of conformal blocks as in [23].

Casimir equation. The quadratic Casimir operator is [26]

$$\widehat{C}_2(r,\eta) = \mathcal{D}_0 + \mathcal{D}_1\,, \tag{B.4}$$

with

$$\mathcal{D}_0 = r^2 \partial_r^2 + (2\nu + 1)\left(\eta \partial_\eta - r\partial_r\right) - \left(1 - \eta^2\right)\partial_\eta^2\,,$$

$$\mathcal{D}_1 = 4r^2 \left[\left(\frac{1 - 2\eta^2 + r^2}{1 + r^4 - 2r^2(2\eta^2 - 1)} - \frac{\nu}{1 - r^2}\right)r\partial_r + \frac{2\eta(1 - \eta^2)}{1 + r^4 - 2r^2(2\eta^2 - 1)}\partial_\eta\right]\,, \tag{B.5}$$

and $\nu = \frac{d-2}{2}$. Using this one finds that

$$\widehat{C}_2(r,\eta)H_\ell^{(d)}(r,\eta) = \lambda_Q H_\ell^{(d)}(r,\eta),$$

$$\lambda_Q = \Delta(\Delta - d) + \ell(\ell + d - 2) - \frac{(d-2)(d-4)r^2}{(1 - r^2)^2} + \frac{4r^2\left(2\eta^2\left(r^4 + 1\right) - \left(r^2 + 1\right)^2\right)}{\left(\left(r^2 + 1\right)^2 - 4\eta^2 r^2\right)^2}. \tag{B.6}$$

Therefore, in the flat-space limit where we send $\Delta \to \infty$ while holding $r, \eta$ fixed, $\lambda_Q \to \Delta^2$ and (B.3) holds.

To fix the normalization factor $\alpha_\ell^{(d)}$ we need to compare the both sides of (B.3) in the limit $z(r,\eta) \ll \bar{z}(r,\eta) \ll 1$, which is translated to $\eta \to \infty, r \to 0$. We find that

$$(G^{\text{pure}})_{\ell+d-1,\Delta-d+1}\left(\rho = r(\eta^2 - \sqrt{\eta^2 - 1}), \bar{\rho} = r^{-1}(\eta^2 + \sqrt{\eta^2 - 1})^{-1}\right) \overset{\eta\to\infty, r\to 0}{\longrightarrow}$$

$$z(r,\eta)^{\frac{d-1-\Delta-\ell}{2}}\bar{z}(r,\eta)^{\frac{\Delta+\ell}{2}} = 2^{d+\ell-1}\eta^{1-d-\ell}r^{d-\Delta-1} \tag{B.7}$$

and therefore

$$\alpha_\ell^{(d)} = \frac{2^{d+2\ell+1}\Gamma\left(\frac{d}{2} + \ell\right)}{\sqrt{\pi}\Gamma\left(\frac{d-1}{2}\right)\Gamma(\ell + 1)}. \tag{B.8}$$

This leads to (5.11) in the main text.

## C   Supplementary details for hyperbolic partial waves

The aim of this section is to derive a limit for the partial waves using the amplitude conjecture (2.3). We will then show that the unitarity condition follows automatically from the CFT axioms. The main arguments of this section were already presented in [21], but here we provide additional details.

### C.1   Large $R$ analysis

Let us first recall that in [21] the correlator is assumed to have the flat-space limit in the region $E'$ defined by $s, t, u < 2m^2$. Applying the conformal dispersion relation to a conformal

correlator of four identical scalars and subtracting the divergences from AdS Landau diagrams we derived a dispersion relation for the flat-space scattering amplitude [21], thus the amplitude is analytic in the cut $s/t$-plane for fixed $4-2\mu_0^2 < u < \mu_0^2$, where $\mu_0$ denotes the mass spectrum gap. By crossing symmetry of the original conformal correlator the amplitude is also analytic in the image regions under crossing.

However, it is obvious that not the entire physical region, e.g. $s > 4, t < 0, u < 0$, is covered, so the dispersion relation is not the ideal tool for proving the partial wave unitarity condition (3.8). Instead, in this section we introduce the "trimmed" correlator using, for example, the $s$-channel OPE

$$\mathcal{G}^{\mathrm{trim}}(s,t,u) = 1 + \sum_{\Delta_{\mathcal{O}} \geq 2\Delta_\phi, \ell_{\mathcal{O}}} a_{\mathcal{O}}^2 G_{\Delta_{\mathcal{O}}, \ell_{\mathcal{O}}}(s,t,u) = \mathcal{G}_{\mathrm{gff}}(s,t,u) + \mathcal{G}_c(s,t,u)\mathcal{T}_R^{\mathrm{trim}}(s,t,u)\,, \tag{C.1}$$

where all the nontrivial conformal blocks below the threshold of *one* OPE channel have been trimmed off. This operation ruins many useful properties such as crossing symmetry, Regge boundedness and the Polyakov-Regge block decomposition, but it preserves the positivity. The trimming also removes divergences caused by the $s$-channel conformal blocks[17], thus the flat-space limit $\mathcal{T}_\infty^{\mathrm{trim}}$ becomes the scattering amplitude in the physical region (this will be justified later)

$$s > 4 : \qquad \mathcal{T}_R^{\mathrm{trim}} \overset{R\to\infty}{\longrightarrow} T(s,t,u)\,. \tag{C.2}$$

The full correlator includes also the disconnected pieces. Therefore

$$\mathcal{G}^{\mathrm{trim}}(s,\eta) - 1 = \left(\frac{4-s}{4-t(\eta)}\right)^{2\Delta_{\mathcal{O}}} + \left(\frac{4-s}{4-u(\eta)}\right)^{2\Delta_{\mathcal{O}}} + \mathcal{G}_c(s,\eta)\mathcal{T}_R^{\mathrm{trim}}(s,\eta) \tag{C.3}$$

must capture the large-$R$ behavior of the correlation function according to the amplitude conjecture. Recall that $t(\eta) = (4-s)(1+\eta)/2$, $u(\eta) = (4-s)(1-\eta)/2$ and $\eta = \cos(\phi)$. The contact diagram in terms of conformal Mandelstam variables reads

$$\mathcal{G}_c(s,t,u) = \frac{w_c 2^{4-4\Delta_\phi}(4-s)^{2\Delta_\phi}}{\sqrt{(4-s)(4-t)(4-u)}}\,, \tag{C.4}$$

Let us now comment on the magnitude of the different terms in equation (C.3) when $\Delta_{\mathcal{O}}$ becomes large. At the leading order we first of all see that the *Euclidean* correlator with $0 < s < 4$ (and real $\phi$) is generally dominated by the disconnected pieces. More precisely, for generic $\phi$ both of the disconnected pieces have a bigger exponential factor than the contact term. And when $\phi = 0$ or $\phi = \pi$ one of the disconnected pieces has the same exponential behavior as the contact term, but the other one will certainly dominate.

---

[17]See discussion above (C.5) on the subtleties related to this subtraction.

There are two subtleties to address. First, the purpose of subtracting conformal blocks below the threshold in (C.1) is to remove divergences caused by the corresponding AdS Landau diagrams. As explained in [15] and Section 7.2, while an AdS Landau diagram has exactly the same exponential growth as a conformal block when it diverges, its divergence region is in fact smaller than the conformal block because the pole corresponding to the AdS Landau diagram is not always picked up (see Figure 4), so outside the region where the AdS Landau diagram diverges we are actually subtracting a divergent contribution from a finite expression and thus $\mathcal{G}^{\mathrm{trim}}$ diverges as $R \to \infty$ when $s \in [0, \mu_0^2]$. Second, the divergences coming from $t$ and $u$-channel AdS Landau diagrams have not been subtracted. This leads to contributions to $\mathcal{G}^{\mathrm{trim}}$ which diverge when $s \in [0, 4 - \mu_0^2]$. However, we have verified that all these divergences are subdominant to the disconnected diagram under our gap assumption $\Delta' < \sqrt{2}\Delta_{\mathcal{O}}$ where $\Delta'$ denotes dimension of exchanged operators. More precisely, after being normalised by the second-sheet contact diagram, these two types of divergences only give subleading contributions (to (C.10) below). So in the Euclidean limit we might as well write:

$$0 < s < 4: \qquad \mathcal{G}^{\mathrm{trim}}(s, \eta) - 1 \overset{R \to \infty}{\longrightarrow} \left( \frac{4 - s}{4 - t(\eta)} \right)^{2\Delta_{\mathcal{O}}} + \left( \frac{4 - s}{4 - u(\eta)} \right)^{2\Delta_{\mathcal{O}}} . \tag{C.5}$$

On the other hand, in the physical kinematics $s > 4$ the exponential factor in the contact diagram is significantly enhanced, and it now *dominates* over the disconnected pieces unless $\phi = 0$ or $\phi = \pi$ where they have the same exponential growth. In this region only $s$-channel Landau diagrams can diverge, but they have been trimmed away.

## C.2   Hyperbolic partial waves

Armed with this insight we can define the objects that will become the partial waves in the flat space-limit. We introduce, in analogy with (3.6), the *hyperbolic partial waves*

$$c_\ell(s) := \frac{\mathcal{N}_d}{2} \int_{-1}^{1} d\eta (1 - \eta^2)^{\frac{d-3}{2}} P_\ell^{(d)}(\eta) \frac{\mathcal{G}(s, \eta) - 1}{\mathcal{G}_c(s, \eta)} . \tag{C.6}$$

Let us now substitute the various building blocks in (C.6). First consider the connected piece. With the decomposition (C.1) and (C.2), we define that

$$s > 4: \qquad f_\ell^{\mathrm{trim}}(s) := \lim_{R \to \infty} \frac{\mathcal{N}_d}{2} \int_{-1}^{1} d\eta (1 - \eta^2)^{\frac{d-3}{2}} P_\ell^{(d)}(\eta) \mathcal{T}_R^{\mathrm{trim}}(r, \eta) . \tag{C.7}$$

More interesting is the computation for the disconnected pieces. In that case the $\eta$-dependence in the exponential factor effectively localizes the $\eta$-integral at $-1$ or $1$, where the

Gegenbauer polynomials produce a factor 1 or $(-1)^\ell$. We find that

$$s > 4: \qquad \frac{\mathcal{N}_d}{2} \int_{-1}^{1} d\eta (1-\eta^2)^{\frac{d-3}{2}} \frac{P_\ell^{(d)}(\eta)}{\mathcal{G}_c(s,\eta)} \left[ \left(\frac{4-s}{4-t}\right)^{2\Delta_\mathcal{O}} + \left(\frac{4-s}{4-u}\right)^{2\Delta_\mathcal{O}} \right] \xrightarrow{R\to\infty}$$

$$-\frac{i}{2} \left( (-1)^\ell + 1 \right) \sqrt{s}(s-4)^{1-d/2} \stackrel{\ell \text{ even}}{=} -i\sqrt{s}(s-4)^{1-d/2}. \quad \text{(C.8)}$$

Altogether we have

$$s > 4: \qquad c_\ell^{\text{trim}}(s) \xrightarrow{R\to\infty} -i\sqrt{s}(s-4)^{1-d/2} + f_\ell^{\text{trim}}(s). \quad \text{(C.9)}$$

To prove unitarity it is useful to introduce the *reflected* hyperbolic partial waves

$$\tilde{c}_\ell(s) := \frac{\mathcal{N}_d}{2} \int_{-1}^{1} d\eta (1-\eta^2)^{\frac{d-3}{2}} P_\ell^{(d)}(\eta) \frac{\mathcal{G}(\tilde{s},\tilde{\eta}) - 1}{e^{-2i\pi\Delta_\mathcal{O}}\mathcal{G}_c(s,\eta)}, \quad \text{(C.10)}$$

together with the involution operation denoted with a tilde

$$(\tilde{s},\tilde{t},\tilde{u}) := (16, -4u, -4t)/s,$$
$$r(\tilde{s}) = -r(s), \quad \eta(\tilde{s},\tilde{t}) = -\eta(s,t) =: \tilde{\eta}. \quad \text{(C.11)}$$

We will always evaluate $\tilde{c}_\ell(s)$ for physical $s > 4$ and $-1 \leq \eta \leq 1$ and so $\mathcal{G}(\tilde{s},\tilde{\eta})$ is always inside the Euclidean triangle. Notice that in both (C.6) and (C.10) the normalisation factor is the contact diagram on the second sheet, which dominates over the disconnected pieces. The extra $e^{-2i\pi\Delta_\mathcal{O}}$ phase is required to remove the rapid oscillation from the contact diagram (coming from $(4-s)^{2\Delta_\mathcal{O}}$ in (C.4)), which was cancelled between the numerator and the denominator in (C.6). Using (C.5) we find that the flat-space limit of $\tilde{c}_\ell^{\text{trim}}(s)$ exists and

$$s > 4: \qquad \tilde{c}_\ell^{\text{trim}}(s) \xrightarrow{R\to\infty} \frac{i}{2} \left( (-1)^\ell + 1 \right) \sqrt{s}(s-4)^{1-d/2} \stackrel{\ell \text{ even}}{=} i\sqrt{s}(s-4)^{1-d/2}. \quad \text{(C.12)}$$

## C.3  Unitarity

Now we would like to demonstrate that any unitary, trimmed conformal correlation function substituted into (C.3) will lead to a scattering amplitude that obeys the unitarity condition (3.8). To do this we would like to show that:

$$s > 4: \qquad |\tilde{c}_\ell(s)| \geq |c_\ell(s)|. \quad \text{(C.13)}$$

Indeed, substituting in (C.9) and (C.12), this condition would yield:

$$s > 4: \qquad \sqrt{s}(s-4)^{1-\frac{d}{2}} \geq \left| -i\sqrt{s}(s-4)^{1-\frac{d}{2}} + f_\ell^{\text{trim}}(s) \right| \quad \text{(C.14)}$$

which proves the existence of $f_\ell^{\text{trim}}(s)$ and more importantly, it is exactly the desired unitarity condition for the amplitude (3.8).

In fact (C.13) does not appear to be easily provable at any finite $R$, but the inequaltiy does emerge easily at very large $R$. To see this, consider first the correlation function $\mathcal{G}(r, \eta)$ itself. Suppose it has a conformal block decomposition of the form

$$\mathcal{G}(r, \eta) - 1 = \sum_{\Delta, \ell} a_{\Delta, \ell} G_{\Delta, \ell}(r, \eta) \tag{C.15}$$

with positive coefficients $a_{\Delta, \ell}$. Now according to [26], the conformal blocks themselves have a radial coordinate expansion of the form

$$G_{\Delta, \ell}(r, \eta) = r^\Delta \sum_{n, \hat{\ell}} B_{\Delta, \ell}^{n, \hat{\ell}} \, r^n P_{\hat{\ell}}^{(d)}(\eta) \tag{C.16}$$

with positive coefficients $B_{\Delta, \ell}^{n, \hat{\ell}}$. So if we write

$$\mathcal{G}(r, \eta) - 1 = \sum_{\ell} k_\ell(r) P_\ell(\eta) \tag{C.17}$$

then $k_\ell(r)$ has a (convergent) $r$ expansion with positive coefficients, and so $k_\ell(r) \geq |k_\ell(-r)|$ for $0 < r < 1$.

Unfortunately the definition of the hyperbolic partial waves includes the division by the contact diagram and therefore the result is not entirely obvious. The main issue is the $\eta$-dependent term in the contact diagram (3.25), which means that it contributes to all the spins in $c_\ell(r)$. Fortunately, at large $R$ this factor cancels because a similar factor arises for each conformal block in $\mathcal{G}(r, \eta)$. The large $\Delta$ limit of a conformal block was given in equation (3.27) which we repeat here:

$$G_{\Delta, \ell}(s, \eta) \overset{\Delta \to \infty}{\longrightarrow} \frac{\sqrt{s} \left( s^{1/4} + \tilde{s}^{1/4} \right)^d}{\sqrt{(4 - t)(4 - u)}} (4r(s))^\Delta P_\ell^{(d)}(\eta) \,. \tag{C.18}$$

Collecting all the $\eta$-independent terms in the contact diagram (C.4) and using the orthogonality of the Gegenbauer polynomials, we find that

$$
\begin{aligned}
c_\ell(s) &\overset{R \to \infty}{\longrightarrow} \frac{\sqrt{s(4 - s)}(s^{\frac{1}{4}} + \tilde{s}^{\frac{1}{4}})^d}{n_\ell^{(d)} w_c 2^{4 - 4\Delta_\mathcal{O}}(4 - s)^{2\Delta_\mathcal{O}}} \sum_{\hat{\Delta}} a_{\hat{\Delta}, \ell}^2 (4r(s))^{\hat{\Delta}} \\
&= i \frac{\sqrt{s(s - 4)}(s^{\frac{1}{4}} + \tilde{s}^{\frac{1}{4}})^d}{n_\ell^{(d)} w_c 2^{4 - 4\Delta_\mathcal{O}}(s - 4)^{2\Delta_\mathcal{O}}} \sum_{\hat{\Delta}} e^{i\pi(\hat{\Delta} - 2\Delta_\mathcal{O})} a_{\hat{\Delta}, \ell}^2 (-4r(s))^{\hat{\Delta}} \,,
\end{aligned}
\tag{C.19}
$$

where the second line is written such that everything apart from $ie^{i\pi(\hat{\Delta} - 2\Delta_\mathcal{O})}$ is real and

positive. The reflected hyperbolic partial wave becomes

$$\tilde{c}_\ell(s) \overset{R\to\infty}{\longrightarrow} i\frac{\sqrt{s(s-4)}(s^{\frac{1}{4}} + \tilde{s}^{\frac{1}{4}})^d}{n_\ell^{(d)}w_c 2^{4-4\Delta_\mathcal{O}}(s-4)^{2\Delta_\mathcal{O}}}\sum_{\hat{\Delta}} a_{\hat{\Delta},\ell}^2(-4r(s))^{\hat{\Delta}}\,. \tag{C.20}$$

Recall $-1 < r(s) < 0$ for $s > 4$, so up to the overall $i$ factor $c_\ell(s)$ is bounded by $\tilde{c}_\ell(s)$ term by term. Thus (C.13) holds in the large $R$ limit and we recover the S-matrix unitarity constraint from conformal unitarity.

### C.3.1 Existence in the physical region

Now that (3.8) has been proved, we can justify the existence of the scattering amplitude in the full physical region. Through the dispersion relation the scattering amplitude exists in the forward scattering limit ($t = 0, \eta = -1$) and its imaginary part is a sum of positive definite terms

$$\operatorname{Im} T(s, \eta = -1) = \sum_\ell n_\ell^{(d)} \operatorname{Im} f_\ell(s) < \infty\,. \tag{C.21}$$

It is then straightforward to show the following bound

$$|T(s,\eta)| \leq \sum_\ell n_\ell^{(d)}|f_\ell(s)| \leq \operatorname{Im} T(s, \eta = -1) \times \max_\ell \frac{|f_\ell(s)|}{\operatorname{Im} f_\ell(s)}\,, \quad s \geq 4\,. \tag{C.22}$$

From (3.8) we can deduce that $\operatorname{Re} f_\ell(s)$ and $\operatorname{Im} f_\ell(s)$ are constrained within a circle $x^2 + (y - \alpha)^2 \leq \alpha^2$ where $\alpha \equiv \sqrt{s}(s-4)^{\frac{2-d}{2}}$. As long as $|f_\ell(s)|, \operatorname{Im} f_\ell(s) \neq 0$, the upper bound in (C.22) is finite. If for fixed $s$ there exists $\ell'$ such that $\operatorname{Im} f_{\ell'}(s) = |f_{\ell'}(s)| = 0$, we can simply exclude $\ell'$ from the sum over spin. Therefore, the unitarity condition together with the existence of $T(s,\eta)$ in the forward scattering limit suffice to establish the existence of $T(s,\eta)$ in the entire physical region.

Let us recap the logic. We define $c_\ell^{\mathrm{trim}}(s)$ as the partial wave projection of $\mathcal{G}^{\mathrm{trim}}$ for any $R$ and show the connected part $f_\ell^{\mathrm{trim}}(s)$ satisfies the unitarity inequality in the limit $R \to \infty$. This leads to the existence of $\mathcal{T}_{R\to\infty}^{\mathrm{trim}}$ in the entire physical region as shown above. There exists an overlap between the dispersion region and the physical region, in which $\mathcal{T}_{R\to\infty}^{\mathrm{trim}} = \mathcal{T}^{\mathrm{sub}}$, because they have the same subtraction in this region. Therefore, $\mathcal{T}_{R\to\infty}^{\mathrm{trim}}$ becomes the same (analytically continued) scattering amplitude as that defined through $\mathcal{T}^{\mathrm{sub}}$ and we conclude that the non-perturbative flat-space amplitude $T(s,\eta)$ exists in the entire physical region.

# D   Saddle point approximation in position space

In this appendix we want to study the flat-space limit of (7.12) which we reproduce here

$$c_{t\text{-exch}}(\Delta, \ell) = 2\kappa_{\Delta+\ell} A_{\hat{\Delta},\Delta_{\mathcal{O}}} \sin^2\left(\pi(\hat{\Delta}/2 - \Delta_{\mathcal{O}})\right) \Omega_{1-h, \frac{\hat{\Delta}}{2}, \Delta_{\mathcal{O}}} \Omega_{\bar{h}, \frac{\hat{\Delta}}{2}, \Delta_{\mathcal{O}}},$$

$$\Omega_{h,h',p} \equiv \int_0^1 \frac{dz}{z^2} \left(\frac{z}{1-z}\right)^p k_{2h}(z) k_{2h'}(1-z),$$
(D.1)

with

$$\Delta = 1 + iRx, \quad h = \frac{1 + iRx - \ell}{2}, \quad \bar{h} = \frac{1 + iRx + \ell}{2}, \quad \hat{\Delta} = \mu R, \quad \Delta_{\mathcal{O}} = mR.$$
(D.2)

Notice that with this parametrisation the two $\Omega$ functions, $\Omega_{\frac{1\pm iRx+\ell}{2}, \frac{\mu R}{2}, mR}$, are related to each other under $x \to -x$. As a result, we only need to analyse one of the $\Omega$'s. Let us denote

$$\Omega(x) \equiv \Omega_{\frac{1+iRx+\ell}{2}, \frac{\mu R}{2}, mR}.$$
(D.3)

For fixed $m, \mu$ and $\ell$, $\Omega(x)$ is regarded as a function of $x = -i\sqrt{s}$. We will assume $\mu > 2m$.

Using the flat-space limit of the $k$ functions [52]

$$k_{2h}(z) = z_2^h F_1(h, h, 2h, z) \overset{|h|\to\infty}{\longrightarrow} \frac{(4\rho(z))^h}{\sqrt{1-\rho(z)^2}}, \quad \rho(z) = \frac{1+\sqrt{1-z}}{1-\sqrt{1-z}},$$
(D.4)

we can rewrite the integral as

$$\Omega(x) \simeq \int_0^1 dz \, g_\Omega(z) \, e^{Rf_\Omega(z)}, \quad R \gg 1,$$
(D.5)

where

$$f_\Omega(z) = \frac{1}{2}(2m + ix)\log(z) + \frac{1}{2}(\mu - 2m)\log(1-z) - ix\log\left(1 + \sqrt{1-z}\right)$$
$$- \mu\log\left(1 + \sqrt{z}\right) + (\mu + ix)\log(2),$$
(D.6)

with two branch cuts $z \in (-\infty, 0]$ and $z \in [1, \infty)$. Let us also record its two analytically

continued version around $z = 0$[18]

$$f_\Omega^{\circlearrowright/\circlearrowleft}(z) = \frac{1}{2}\left(2m + ix\right)\left(\log(z) \pm 2\pi i\right) + \frac{1}{2}(\mu - 2m)\log(1-z) - ix\log\left(1 + \sqrt{1-z}\right)$$
$$- \mu\log\left(1 - \sqrt{z}\right) + (\mu + ix)\log(2),$$
(D.8)

which will be useful later. The $O(1)$ factor is

$$g_\Omega^{(\ell)}(z) = 2^{\ell-1}\frac{(1 + \sqrt{z})}{(1-z)^{1/4}z^{7/4}}\left(\frac{1 - \sqrt{1-z}}{1 + \sqrt{1-z}}\right)^{\ell/2}.$$
(D.9)

The saddle points are

$$z^\pm(x) = \left(\frac{2\mu m \pm \sqrt{x^2\left(-\mu^2 + 4m^2 + x^2\right)}}{x^2 - \mu^2}\right)^2,$$
(D.10)

and in particular we have

$$z^-(\pm i2m) = 0, \qquad \lim_{|x|\to\infty} z^\pm(x) = 1.$$
(D.11)

These points are interesting because the potential divergences of (D.1) come from the regions near $z = 0, 1$. To have more intuition about the saddle points we can use conformal Mandelstam variable relations to find

$$s(z = z^\pm, \bar{z} = z^\mp) = -x^2 = s, \qquad\qquad t(z = z^\pm, \bar{z} = z^\mp) = \mu^2,$$
(D.12)

which indicates these saddle points, with the proper combination, indeed localise the integrals at the expected locations.

In a careful saddle point analysis, one has to check whether the original integration contour can be deformed into a combination of steepest descent contours going through the saddle point(s) and the integration endpoints,[19] and check whether there are poles picked up

---

[18]The careful reader may worry that the continued version of (D.4) does not remain a good approximation to $k_{2h}(z)$ on the secondary sheets. The worry is valid. In the following we will first only need to consider continuation around $z = 0$ (see Figure 11). Then this worry does not matter, because the relevant function is $k_{\mu R}(1-z)$, where $\mu$ is the mass of the exchanged particle. More concretely, one can first work out the discontinuity of the hypergeometric $_2F_1$ function

$$\mathrm{Disc}[_2F_1(h, h, 2h; 1-z)] = \mp 2\pi i\frac{\Gamma(2h)}{\Gamma(h)^2}\,_2F_1(h, h, 1; z), \quad z \to ze^{\pm 2\pi i},$$
(D.7)

and use, for example, saddle point analysis to work out the large-$h$ approximation of the discontinuity. The continued $_2F_1$ contains two terms which scale as $(1 - \sqrt{z})^{-\mu R}$ and $(1 + \sqrt{z})^{-\mu R}$ respectively, and the first term appears on the second line of (D.8). Since $\mu R > 0$, one finds that the first term always dominates over the second on the entire complex-$z$ plane, hence (D.8) suffices. As for the continuation around $z = 1$, we need to consider $k_{\pm ixR}(z)$ with $x \in \mathbb{R}$, then the dominance can shift between the two terms and this is the Stokes phenomenon. We will discuss this in Section D.2.

[19]For discussion on endpoints see Section 6.

during the contour deformation. Below we will first focus on one of the saddle points and discuss the subtleties later.

The main takeaway is that in order to capture the flat-space limit of (D.1), we need to use all three forms of the large exponent $f_\Omega$ (D.6) and (D.8), depending on which region of complex $x$-plane we are at. The prescription is given in Figure 11. The divergence region comes from the switch of dominance (the Stokes phenomenon) of the two saddle points in (D.10).

## D.1 Contribution from the saddle point

The saddle point approximation turns out to be rather involved. This is mainly because $f_\Omega(z)$ contains two branch cuts in $z$. The saddle points (D.10) are functions of $x$, and as $x$ moves on the complex plane the saddle points can go into different branches of $f_\Omega(z)$.

To determine which branch of $f_\Omega(z)$ and which saddle point(s) to use, we start from a point close to the origin on the complex-$x$ plane, say $x = 0.1 + 0.1i$, which corresponds the vicinity of $\Delta = d/2$ and is away from various singularities or the bad region seen in section 7.5.2. Next we use the numerics in 7.5.2 to verify that in this region we should use $f_\Omega(z)$ on the principal sheet and the saddle point $z^+(x)$ from (D.10). After that we can analytically continue accordingly as $x$ moves on the complex plane. Figure 11 dictates how we should switch saddle points and branches of $f_\Omega(z)$ as $x$ moves on its complex plane.[20]

Through numerical checks we find that there are three half lines in on the complex-$x$ plane that we should not analytically continue across in Figure 11 (one black and two orange). The black half line also corresponds to the condensing physical poles in $c_{t\text{-ex}}(\Delta, \ell)$. The flat-space limit of $\Omega$ is not well defined there, but the saddle point is. This is because the physical singularity originates from the vicinity of $z = 0$ and the saddle point is in general away from that point. When $x = \pm 2im$ the saddle points coincide with $z = 0$ and the saddle point approximation diverges, reproducing the pole at $s = 4m^2$ in (7.2). Through $x = -i\sqrt{s}$, the orange half lines should become the left branch cut of $f_\ell(s)$ in the flat-space limit. As shown in Figure 11, to cover the entire complex $x$-plane we need to use the exponent function $f_\Omega(z)$ (in the white region) as well as its two analytical continuations around $z = 0$ (in the blue and orange region). This was anticipated around (D.8). With respect to the branch cut $z \in [1, \infty)$ we only use $f_\Omega(z)$ in the principal branch.

With the procedure explained, now the saddle point approximation reads

$$\Omega(x) \simeq \sqrt{\frac{2\pi}{e^{-i\pi}R(f_\Omega^{(\circlearrowleft/\circlearrowright)})''(z^\pm(x))}}\, g_\Omega^{(\ell)}(z^\pm(x)) e^{R f_\Omega^{(\circlearrowleft/\circlearrowright)}(z^\pm(x))}, \quad R \gg 1, \tag{D.13}$$

where the explicit choices of the saddle point and $f_\Omega$ are dictated in Figure 11. The contri-

---

[20] In this way we always use only one of the saddle points, but we will see that this does not always capture the full integral.

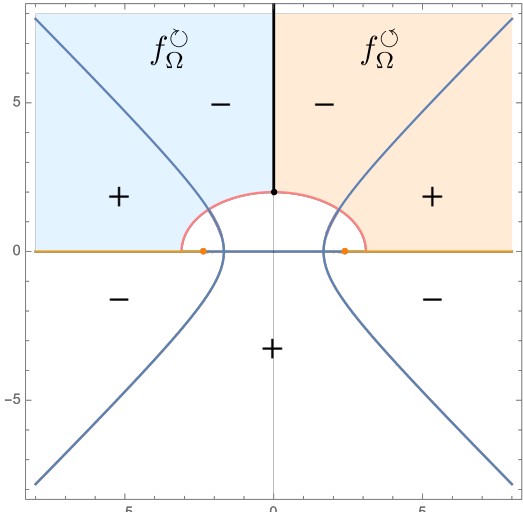

**Figure 11**. Complex-$x$ plane with $\mu = 3.1, m = 1$ for $\Omega(x)$ as $R \to \infty$. The orange lines and the pink curve are determined by setting the saddle points on the branch cuts of $f_\Omega(z)$, $z^\pm(x) \in [1, \infty)$ and $z^\pm(x) \in (-\infty, 0]$, respectively. The pink curve is the upper half of the ellipse $x^2/\mu^2 + y^2/4m^2 = 1$ and the orange dots at $x = \pm\sqrt{\mu^2 - 4m^2}$ are both the endpoints of the orange lines and the foci of the pink ellipse. The orange dots and lines are the expected branch point and cut in $f_\ell(s)$. The thick black line starting from $x = i2m$ indicates the physical poles of $c(\Delta, \ell)$ condensing in the flat-space limit (recall $\Delta - d/2 = iRx$ so there is a 90 degree rotation between $\Delta$-plane and $x$-plane). The blue curves correspond to the square root branch cut in the saddle point solution (D.10), across which the two saddle points are exchanged.

How to use the plot: We should use $f_\Omega(z)$ in the principal-sheet in the white region and $x$ going through the pink curve from the white region to the blue (orange) region corresponds to the saddle point going through the branch cut $z \in (-\infty, 0]$ in the (counter-)clockwise direction. The orange and the black branch cuts should not be crossed. The "$\pm$" signs indicate which saddle point in (D.10) to use. For $\Omega(-x)$ we simply rotate the figure by 180 degrees.

bution to (D.1) from factors other than the $\Omega$ functions is

$$2\kappa_{1+iRx+\ell} A_{\hat{\Delta}, \Delta_\mathcal{O}} \sin^2\left(\frac{\mu R - 2mR}{2}\pi\right) \simeq g_0 e^{R f_0(x)}, \quad R \gg 1, \mu > 2m,$$

$$f_0(x) = 2m \log\left(\frac{\mu^2 - 4m^2}{4m^2}\right) + \mu \log\left(\frac{\mu + 2m}{4\mu - 8m}\right) - 2ix \log(2), \quad \text{(D.14)}$$

$$g_0^{(\ell)} = \frac{4^{-\ell-1} m^2}{\pi^3 \mu(\mu - 2m)(\mu + 2m)^3}.$$

Finally, combining all the components in (D.1) we find that

$$c_{t\text{-exch}}(\Delta = 1 + iRx, \ell) \simeq g_{t\text{-exch}}^{(\ell)}(x) e^{R f_{t\text{-exch}}(x)}, \quad R \gg 1, \quad \text{(D.15)}$$

with

$$f_{t\text{-exch}}(x) = f_0(x) + f_\Omega^{(\circlearrowright/\circlearrowleft)}(z^\pm(x)) + f_\Omega^{(\circlearrowright/\circlearrowleft)}(z^\mp(-x)),$$

$$g_{t\text{-exch}}^{(\ell)}(x) = -\frac{2\pi g_0^{(\ell)}}{R}\left(\sqrt{\frac{1}{(f_\Omega^{\circlearrowright/\circlearrowleft})''(z^\pm(x))}}\,g_\Omega^{(\ell)}(z^\pm(x))\right)\left(\sqrt{\frac{1}{(f_\Omega^{\circlearrowright/\circlearrowleft})''(z^\mp(-x))}}\,g_\Omega^{(\ell)}(z^\mp(-x))\right),$$

$$\text{(D.16)}$$

where again the explicit choices of the saddle point and $f_\Omega$ are dictated in Figure 11. In particular, from the figure we see that $\Omega(x)$ and $\Omega(-x)$ always use different saddle points in (D.10), thus the "proper combination" to obtain (D.12) is always guaranteed.

It can be checked that in different regions of the complex-$x$ plane, when the saddle point and the branch of $f_\Omega$ are correctly chosen, $f_{t\text{-exch}}(x)$ agrees with the large exponent of contact diagram's OPE density, $f_c(x)$:

$$f_{t\text{-exch}}(x) = f_c(x), \tag{D.17}$$

where for the OPE density of contact diagram given in (3.17) (with all external dimensions set equal) we have

$$c_c(\Delta = 1 + iRx, 0) \simeq g_c(x)\,e^{Rf_c(x)}, \quad R \gg 1,$$

$$g_c(x) = \frac{m^2}{8\pi^2 R\left(m - \frac{ix}{2}\right)\left(m + \frac{ix}{2}\right)(4m^2 + x^2)},$$

$$f_c(x) = (2m + ix)\log(2m + ix) + (2m - ix)\log(2m - ix) - 4m\log(2m) - 2ix\log(2). \tag{D.18}$$

This shows that although $f_0(x)$ and $f_\Omega(x)$ have explicit $\mu$-dependence, $f_{t\text{-exch}}(x)$ is independent of $\mu$, which is in consistency with the amplitude conjecture.

For the $O(1)$ factor $g_{t\text{-exch}}(x)$ one can check that

$$\frac{g_{t\text{-exch}}^{(\ell)}(x(s))}{g_c(x(s))} = n_\ell^{(d)} f_\ell(s), \tag{D.19}$$

which is consistent with (3.35) but with a notable difference: (D.19) *works everywhere* while (3.35) fails in the "bad" region. In particular, the branch point in the $f_\ell(s)$ is reproduced by the zero in $f_\Omega''(z^+(x))$ and $f_\Omega''(z^-(-x))$. By expanding both sides of (D.19) around $x =$

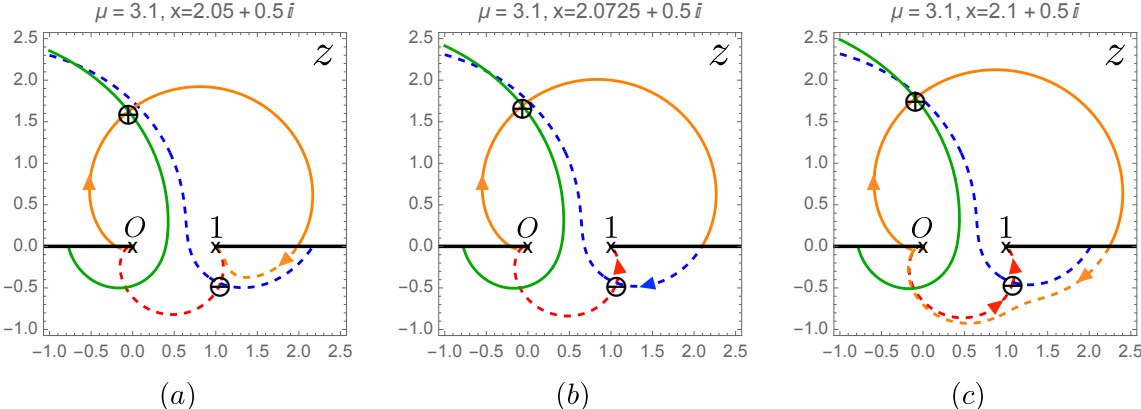

**Figure 12**. Crossing the Stokes line. Orange (red) and green (blue) curve are the steepest descent and ascent contour through the "+" ("-") saddle point. Dashed curves are on the second sheet around $z = 1$. The arrows indicate the integration direction. ($a$): before crossing the Stokes line only the "+" saddle point is needed to approximate the original integral on $z \in [0, 1]$. ($b$): when $x$ is on the Stokes line the two saddle points are connected by the same constant-phase contour. Note that this contour is the steepest *ascent* contour for the "-" saddle point, so its magnitude is much smaller than that of the "+" saddle point. The integral first passes through the "+" saddle point, then hits the "-" saddle point and finally follows the red curve to $z = 1$. ($c$): after passing the Stokes line both saddle points are needed to approximation the original integral. Once the Stokes line is crossed, the magnitude of "-" saddle point does not have to be smaller than the "+" saddle point.

$\sqrt{\mu^2 - 4m^2}$ we find the same singular behaviour[21]

$$\frac{g_{t\text{-exch}}(x(s))}{g_c(x(s))} = n_\ell^{(d)} f_\ell(s) = O\left(\left(x - \sqrt{\mu^2 - 4m^2}\right)^{-1/2}\right), \quad x \to \sqrt{\mu^2 - 4m^2}, x < \sqrt{\mu^2 - 4m^2}.$$
(D.20)

Furthermore, as illustrated in Figure 11, the branch cut $z \in [1, \infty)$ in the integrand of $\Omega$ is mapped to the branch cut $x \in [\sqrt{\mu^2 - 4m^2}, \infty)$ of the flat-space limit of $\Omega$. This is reminiscent of the case in Section 7.1.

## D.2   Origin of the bad region

We have seen that the contribution solely from one of the saddle points of the inversion integral (D.1) exactly reproduces the flat space as shown in (D.17) and (D.19). However, exact results in Figure 7 indicate that the saddle point does not capture everything. Indeed, as mentioned previously a careful saddle point analysis requires one to examine the entire contour deformation, which we will now discuss.

It turns out that the other saddle point neglected in the previous discussion[22] is key reason

---

[21]Here we see a square root branch cut because when $d = 2$ the Gegenbauer $Q$-function in (7.2) does not have a branch cut.

[22]When crossing the blue curves in Figure 11 the two saddle points switch, but previously we considered only one of the saddle points each time.

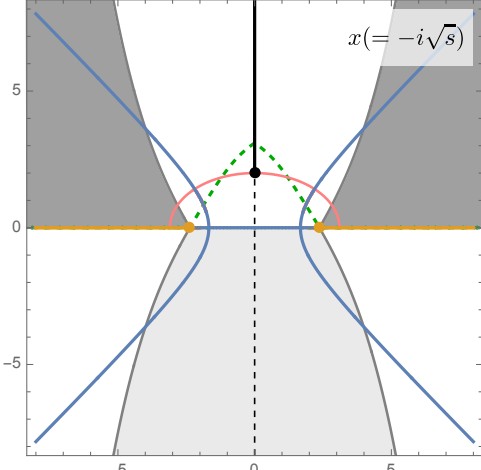

**Figure 13**. Stokes and anti-Stokes lines on the complex-$x$ plane with $\mu = 3.1m$. The green dashed curve is the Stokes line. (Its image under reflection with respect to the real axis is also Stokes line but it induces the opposite steepest descent direction to that in Figure 12 and is irrelevant.) Starting from the triangular region near the origin, one has to include both saddle points after crossing the Stokes line. The gray curves are the anti-Stokes lines on which the two saddle point has the same exponential magnitude. The unwanted saddle point dominates in the gray regions. Only the darker gray regions are the bad regions because $x$ does not cross the Stokes line to reach the light gray regions. (Recall the orange lines should not be crossed.) The darker gray regions match that in Figure 7.

for the presence of the bad region, and it joins the game through the Stokes phenomenon (also see footnote 18). When $x$ is continued away from the origin in Figure 11, it can cross the Stokes line[23] and then one has to pick up both saddle points to have a good approximation to the original integral. This is illustrated in Figure 12.

However, picking up the second saddle point does not necessarily make the previous saddle point approximation (D.15) invalid. One still needs to compare the magnitude of the two saddle points, the equivalence of which determines the anti-Stokes line. Similar to the previous analysis in Section D.1, here we also need to determine which branch of $f_\Omega$ to use for the other saddle point. After that, we can identify numerically the Stokes and the anti-Stokes lines and carve out the bad region. The result of an example with $\mu = 3.1m$ is shown in Figure 13, where the *darker* gray regions are the bad regions, which are very similar in shape to that in Figure 7. In the lighter shaded region the second saddle point also dominates but it is not picked up. Note that Figure 13 shows the result for $\Omega(x)$ only. To match with Figure 7 we also need to include the bad regions for $\Omega(-x)$, for which we only need to rotate Figure 13 by 180 degrees. By grouping together the results for $\Omega(x)$ and $\Omega(-x)$ we simply get the upper half of Figure 13 and its image under reflection with respect to the real axis. The full results of the bad region on complex-$s$ plane for various values of $\mu$ are shown in Figure 6.

---

[23]We use the convention that on the Stokes line the integrand at the two saddle points have the same phase, and the same magnitude on the anti-Stokes line.

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
