# Peer review of "Flat-space Partial Waves From Conformal OPE Densities"

_SciPost Physics_

## Round 2 · Referee Report · Anonymous (Referee 1) · 2024-5-9

Strengths

  1. The paper proposes a formula to relate the OPE data to partial wave coefficients in the flat-space limit in certain analytic regions. This can be more generally understood as relations between large scaling dimension sections of CFTs and flat-space scattering data.

  2. The paper elegantly shows that, when avoiding subtle analytic regions, the flat-space limit of the Euclidean inversion formula and the Lorentzian inversion formula of CFTs can indeed recover the standard partial wave projections and the Froissart-Gribov formula of flat-space scattering amplitudes, respectively.

  3. The paper also elegantly demonstrates that the flat-space limit of the conformal dispersion relation can also reproduce the S-matrix dispersion relation.

  4. Not only providing strong evidence for the proposal, the paper also faithfully reports the subtlety where the flat-space limit breaks down, suggesting interesting, important but unknown analytic properties of the flat-space limit of conformal correlators.

  5. The paper paves the way for excellent routes to understand the analytic properties of the S-matrix from conformal field theories.

  6. The paper is well organized and conveys the main points clearly.

Weaknesses

The paper is excellent and enjoyable for experts to read; however, it may not be very friendly for non-experts with general knowledge of conformal field theories and S-matrix to gain insights into this subject. The essential reason is that the paper is full of technical details but lacks a focus on the physical picture and interpretation; probably the authors assume the readers are fully knowledgeable about the physics behind the subject.

Report

The paper considers the conformal partial wave expansion of the conformal four-point function and studies its flat-space limit where all scaling dimensions are large. This limit is expected to provide insights into the massive S-matrix in flat space. Based on supportive examples, the authors conjecture a beautiful dictionary to relate the scaling limit of OPE data to the partial wave coefficients. Moreover, as naively expected, the authors indeed show that the inversion formulas and dispersion relations of the S-matrix can be obtained by taking the flat-space limit of their counterparts in CFTs. The paper also identifies the analytic domains where these relations may break down. The paper is generally elegant and is important for improving our understanding of S-matrix analyticity from CFTs, where the analyticity is more rigorously established.

The contents and scientific values of this paper definitely qualify it for SciPost. I recommend its publication in SciPost, but with the slight modifications noted below in the "requested changes".

Requested changes

  1. I believe that the picture of stable states as poles and resonances as complex poles (in the second sheet) in either OPE or partial waves should be explained in more details. The OPE data encodes the physical operators as poles OPE^2/(\Delta-\Delta_{phys}). If it gives rise to partial wave coefficients, these poles become poles in s like 1/(s-s0). However, I'm confused about the scenarios with resonances. Generally, resonances should be complex poles of f_{\ell} in the second sheet, however, how to make sense of branch cuts and complex poles in OPE density?

  2. The Lorentzian inversion formula and Froissart-Gribov formula analytically continue the data in spin. One of the reasons we want to do this, for example, in the S-matrix, is that it helps to analytically continue the S-matrix to extreme kinematics like the Regge limit and ensures the convergence. Consider the t-channel Regge limit; the result is dominated by the leading Regge intercept defined at s -> 0; in CFT, the Regge intercept is also precisely \Delta -> d/2. However, this then corresponds to finite scaling dimensions. This means that all the analysis with the flat-space limit just badly breaks down here. This is confusing because it means that the Regge limit is not commutable with the flat-space limit. This might be beyond the scope of this paper, but it would be great if the authors could provide small comments on this point; otherwise, I'm not sure why we need to analytically continue in spin if these effects can never be captured by the flat-space limit.

3.In the proof of the flat-space limit of the Lorentzian inversion formula, the authors showed that one pure power law part of the inverted conformal block reduces to the Q-function in (5.11), while stating that another part does not contribute to the saddle point (above (5.12)). This sounds specific to the conformal frame that the authors chose. In 1711.03816, the Lorentzian inversion formula can be put into a form with a double commutator multiplied by the exact Q-function, see (3.24) there. So, we can keep the whole block here, while the double commutator plays a role like a discontinuity. This is puzzling because it looks like the saddle point of the conformal block is gauge dependent: block at saddle in one frame = block at saddle in another frame + small. I thus believe that the authors should elaborate a bit on the gauge choice in the saddle point analysis.

  1. Perhaps the authors can elaborate a bit more on reviewing the dangerous regions in s of the flat-space limit of both the amplitude conjecture and the partial wave conjecture? What I mean is that it would be great to see how badly they diverge when the flat-space limit does not work. For example, if we expect that the divergence comes from an AdS Landau diagram, similar to a flat-space Landau diagram, there should exist a clear pinch that leads to IR or collinear divergence like Log[R_{AdS}^2/s], right? In flat-space, whenever there are unavoidable pinches, the behavior of the divergence actually signals nontrivial physics.

  2. Very small typos: In (4.4) and other places like (4.7), the notation for spin is not unified; both l and \ell are used to represent spin.

Recommendation

Ask for minor revision

---

## Round 2 · Referee Report · Anonymous (Referee 2) · 2024-5-14

Report

This paper adds to the literature on how to connect flat space scattering amplitudes to CFT correlators by taking the flat space limit of AdS/CFT. The main focus of the paper is to investigate the validity of what the authors call their "partial wave conjecture", which is that the flat space partial waves are a limit of the connected CFT OPE densities. They are particularly concerned with its validity upon complexification of the arguments, and what regions in the covering space of the complex plane it does or does not hold. There are a number of detailed worked examples, as well as some nice results on how various flat space formulas emerge from analogous CFT formulas, and the discussion of some of the subtleties involved in using the formula upon complexification are likely to be helpful to future investigations.

I have a few comments and questions for the authors.

  1. I unfortunately did not come away with a clear sense of a characterization of when the formula works or not, although the authors do frequently warn about the dangers of exchanging orders of limits when saddle points are involved and discuss specific regions where the saddle point approximation they need to use in order to derive their relation cannot be justified. On the other hand, they also provide examples where these subtleties apparently do not affect the final result. The upshot seems to be that they provide partial guidance to the reader for where challenges might arise. Moreover, they claim to prove that their relation is never valid in the complex plane in a region near the `left cut' of the amplitude, i.e. the cut at $s< 4m^2 - t_0$ that arises due to crossing symmetry. So the status seems to be that they have a formula that they know cannot be correct in the full complex plane, which is what I thought the goal was, and they have at best partial knowledge about when it is valid. Is this a negative result that indicates their relation (1.2) is not what one wants to use for studying analytic properties of flat-space scattering using holography? Or are they proposing that one should still use this relation, prove that it is valid in some regions, and then prove analytic S-matrix properties in those regions? If it is the latter, what do they think the realistic goals are? If not, what do they envision as possible improvements to relation (1.2)?

  2. I would naively have thought that their relation (1.2) suffers from a fundamental obstacle from the start as a relation that could be fully valid upon complexification and analytic continuation of both sides, because the analytic structure of the two sides is manifestly different. That is, the OPE densities have poles, whereas the flat-space scattering amplitudes have multiple overlapping cuts (from the multi-particle thresholds at $s=(n m)^2$ for $n=2,3,\dots$. ). If the analytic continuation is taken before the flat space limit, then the analytic continuation of the OPE densities will not pass through these cuts to the second and higher sheets, whereas if the analytic continuation is taken after taking the flat-space limit then that would seem to defeat the authors' purpose. Why is this not a problem?

  3. Relatedly, the authors consider two example calculations, a contact diagram and a tree-level exchange. However, since the point of the paper is to understand the analytic structure of (1.2), these two examples do not really seem to get at the heart of what they are trying to explore, which involves the complicated analytic structure of amplitudes that first begins to show up at loop level. I suspect that adding a loop computation example to their paper at this point might seem to them like a fairly onerous burden just to merit publication. But it seems to me that such an example would be a particularly useful addition to the paper, as well as an illuminating and more direct test of their relation. I also do not think it would be very hard. Using the relation (53) from arXiv:1111.6972, a simple bulk loop diagram can be reduced to a sum over tree-level scalar exchanges with known coefficieients: \begin{equation} c_{s-{\rm loop}}(\Delta,0) = \sum_{n=0}^\infty N_{\Delta_\chi}(n) c_{s-{\rm exch}}(\Delta,0;\Delta_b = 2\Delta_\chi+2n) \end{equation} where $\Delta_\chi$ is the dimension of the scalar running around the loop, and the coefficients are \begin{equation} N_{\Delta_\chi}(n) = \frac{(\frac{d}{2})n (2\Delta\chi + 2n){1-\frac{d}{2}} (2\Delta\chi+n - d+1)n}{2 \pi^{d/2} n! (\Delta\chi+n)^2_{1-\frac{d}{2}} (2\Delta_\chi+n-\frac{d}{2})n}, \end{equation} and $c_{s-{\rm exch}}$ is given by (4.1) in the authors' paper, \begin{equation} c c_c(\Delta,0). \end{equation} Thus the calculation is straightforward. Moreover, for }}(\Delta, 0; \Delta_b = 2\Delta_\chi+2n) = \frac{-R^2}{(\Delta-d/2)^2 - (2\Delta_\chi+2n-d/2)^2$d\le 2$, the bulk interaction is relevant and so the sum on $n$ converges without needing any regularization and renormalization. The analytic structure in $\Delta$ at finite $R$ is manifest in such a formula. Would the authors consider adding this calculation and discussing its implications? As mentioned in the previous point (2) above, this analytic structure would not seem to permit the type of analytic structure that emerges in the flat-space limit, with many branch cuts and multi-sheeted covering spaces, and I thought it was exactly this analytic continuation of flat-space amplitudes that the authors hoped to say something about.

Incidentally, a relation between partial wave scattering amplitudes and OPE coefficients that is quite similar to that of the authors appears in arXiv:1111.6972 equation (71), and perhaps the authors could comment on how their conjecture (1.2) relates to it.

  1. The authors claim that their relation does not hold when applied to the disconnected part of the correlator, and in fact that the necessary limit does not even exist. However, if they want their relation to be nonperturbative, then I believe this means they will also face the same problem in the connected piece, because at a nonperturbative level the connected piece of the partial wave amplitude contains a contribution proportional to the disconnected piece.
    As is evident from partial wave unitarity, if one only looks at a single partial wave at a time then the connected and disconnected pieces are intimately related and there is not a clean separation between them (there would be in momentum space, or if one considers the asymptotic large $\ell$ limit of partial waves). So I believe that they would eventually find the same problem they identify in the disconnected piece showing up in the connected piece as well. This is manifest for instance in the one-loop example mentioned above, where the sum can be done in closed form when $d=2$. On the other hand, they also indicate that the problem is fairly minor and can be resolved by simply performing some smearing over $s$ (probably with a principle value prescription to accommodate the condensing line of poles). Do the authors agree that their problem with the disconnected piece also exists in the connected piece? If not, why not, and if so, do they agree it can be resolved by doing some averaging, or some other procedure they might have in mind?

Recommendation

Ask for minor revision

---

## Editorial Decision

awaiting_resubmission